# The development of *Nanosota-1* as anti-SARS-CoV-2 nanobody drug candidates

Gang Ye[1,2†], Joseph Gallant[3†], Jian Zheng[4†], Christopher Massey[5], Ke Shi[6], Wanbo Tai[7], Abby Odle[4], Molly Vickers[4], Jian Shang[1,2], Yushun Wan[1,2], Lanying Du[7], Hideki Aihara[6], Stanley Perlman[4]\*, Aaron LeBeau[3]\*, Fang Li[1,2]\*

[1]Department of Veterinary and Biomedical Sciences, University of Minnesota, Saint Paul, United States; [2]Center for Coronavirus Research, University of Minnesota, Saint Paul, United States; [3]Department of Pharmacology, University of Minnesota, Minneapolis, United States; [4]Department of Microbiology and Immunology, University of Iowa, Iowa City, United States; [5]Institutional Office of Regulated Nonclinical Studies, University of Texas Medical Branch, Galveston, United States; [6]Department of Biochemistry, Molecular Biology and Biophysics, University of Minnesota, Minneapolis, United States; [7]Laboratory of Viral Immunology, Lindsley F. Kimball Research Institute, New York Blood Center, New York, United States

**\*For correspondence:**
stanley-perlman@uiowa.edu (SP);
alebeau@umn.edu (ALB);
lifang@umn.edu (FL)

†These authors contributed equally to this work

**Abstract** Combating the COVID-19 pandemic requires potent and low-cost therapeutics. We identified a series of single-domain antibodies (i.e., nanobody), *Nanosota-1*, from a camelid nanobody phage display library. Structural data showed that *Nanosota-1* bound to the oft-hidden receptor-binding domain (RBD) of SARS-CoV-2 spike protein, blocking viral receptor angiotensin-converting enzyme 2 (ACE2). The lead drug candidate possessing an Fc tag (*Nanosota-1C-Fc*) bound to SARS-CoV-2 RBD ~3000 times more tightly than ACE2 did and inhibited SARS-CoV-2 pseudovirus ~160 times more efficiently than ACE2 did. Administered at a single dose, *Nanosota-1C-Fc* demonstrated preventive and therapeutic efficacy against live SARS-CoV-2 infection in both hamster and mouse models. Unlike conventional antibodies, *Nanosota-1C-Fc* was produced at high yields in bacteria and had exceptional thermostability. Pharmacokinetic analysis of *Nanosota-1C-Fc* documented an excellent in vivo stability and a high tissue bioavailability. As effective and inexpensive drug candidates, *Nanosota-1* may contribute to the battle against COVID-19.

## Introduction

The novel coronavirus SARS-CoV-2 has led to the COVID-19 pandemic, devastating human health and global economy (*Li et al., 2020*; *Huang et al., 2020*). Anti-SARS-CoV-2 drugs are urgently needed to treat patients, save lives, and help revive economy. Yet daunting challenges confront the development of such drugs. Though small-molecule drugs could target SARS-CoV-2, it can take years to develop them and their use is often limited by poor specificity and off-target effects. Repurposed drugs, developed against other viruses, also have low specificity against SARS-CoV-2. Therapeutic antibodies have been identified and generally have high specificity; however, their expression in mammalian cells often leads to low yields and high production costs (*Salazar et al., 2017*; *Breedveld, 2000*). A realistic therapeutic solution to COVID-19 must be potent and specific, yet easy to produce.

Nanobodies (or VHH antibodies) are unique antibodies derived from heavy-chain-only antibodies found in members of the camelidae family (llamas, alpacas, camels, and so on) (*Figure 1*; *Figure 1—figure-supplement 1*; *Könning et al., 2017*; *De Meyer et al., 2014*). Because of their small size (2.5 nm by 4 nm; 12–15 kDa) and unique binding domains, nanobodies offer many advantages over

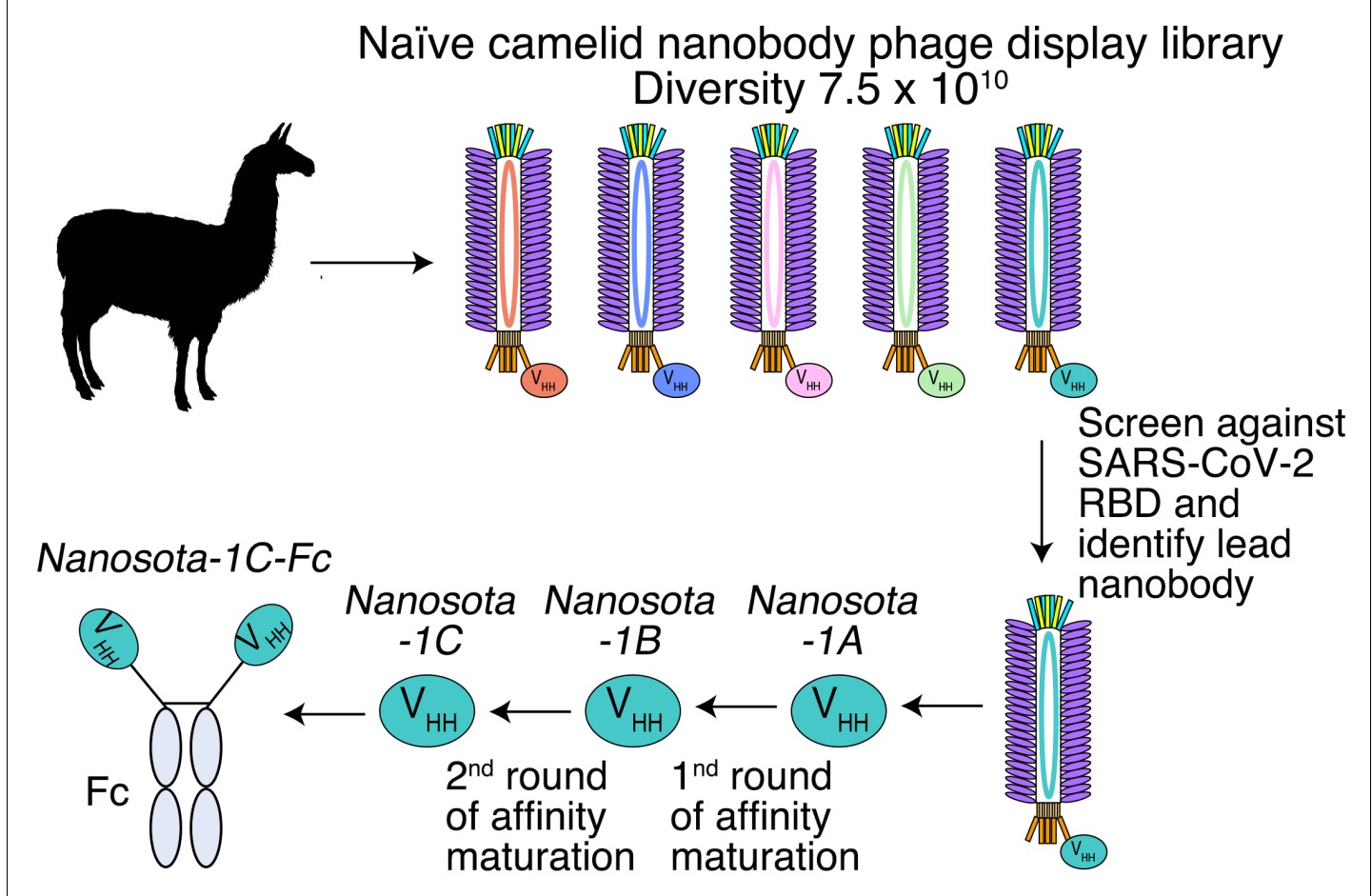

**Figure 1.** Construction of a camelid nanobody phage display library and use of this library for screening of anti-SARS-CoV-2 nanobodies. A large-sized (diversity 7.5 x $10^{10}$) naive nanobody phage display library was constructed using B cells of over a dozen llamas and alpacas. Phages were screened for their high binding affinity for SARS-CoV-2 receptor-binding domain (RBD). Nanobodies expressed from the selected phages were further screened for their potency in neutralizing SARS-CoV-2 pseudovirus entry. The best performing nanobody was subjected to two rounds of affinity maturation. The online version of this article includes the following figure supplement(s) for figure 1:

**Figure supplement 1.** Schematic drawings of nanobodies and conventional antibodies.

conventional antibodies including the ability to bind cryptic epitopes on their antigen, high tissue permeability, ease of production, and thermostability (*Muyldermans, 2013*; *Steeland et al., 2016*). Although small, nanobodies bind their targets with high affinity and specificity due to an extended antigen-binding region (*Muyldermans, 2013*; *Steeland et al., 2016*). One drawback of nanobodies is their quick clearance by kidneys due to their small size; this can be overcome by adding tags to increase the molecular weight to a desired level that is above the kidney clearance threshold but still much lower than conventional antibodies' molecular weight. Underscoring the potency and safety of nanobodies as human therapeutics, a nanobody drug was recently approved for clinical use in treating a blood clotting disorder (*Scully et al., 2019*). Additionally, due to their superior stability, nanobodies can be inhaled to treat lung diseases (*Van Heeke et al., 2017*) or ingested to treat intestine diseases (*Vega et al., 2013*). Nanobodies are currently being developed against SARS-CoV-2 to combat COVID-19 (*Huo et al., 2020*; *Hanke et al., 2020*; *Xiang et al., 2020*; *Schoof et al., 2020*; *Wrapp et al., 2020a*). However, these reported nanobodies have not been adequately evaluated for therapeutic efficacy in vivo.

The receptor-binding domain (RBD) of the SARS-CoV-2 spike protein is a prime target for therapeutic development (*Li, 2015*). The spike protein guides coronavirus entry into host cells by first binding to a receptor on the host cell surface and then fusing the viral and host membranes

(*Li, 2016*; *Perlman and Netland, 2009*). The RBDs of SARS-CoV-2 and a closely related SARS-CoV-1 both recognize human angiotensin-converting enzyme 2 (ACE2) as their receptor (*Li, 2015*; *Wan et al., 2020*; *Li et al., 2003*; *Zhou et al., 2020*). Previously, we have shown that SARS-CoV-1 and SARS-CoV-2 RBDs both contain a core structure and a receptor-binding motif (RBM) and that SARS-CoV-2 RBD has significantly higher ACE2-binding affinity than SARS-CoV-1 RBD due to several structural changes in the RBM (*Shang et al., 2020a*; *Li et al., 2005*). We have further shown that SARS-CoV-2 RBD is more hidden than SARS-CoV-1 RBD in the entire spike protein as a possible viral strategy for immune evasion (*Shang et al., 2020b*). Hence, to block SARS-CoV-2 binding to ACE2, a nanobody drug would need to bind to SARS-CoV-2 RBD more tightly than ACE2.

Here we report the development of a series of anti-SARS-CoV-2 nanobody drug candidates, *Nanosota-1*. Identified by screening a camelid nanobody phage display library against the SARS-CoV-2 RBD, the *Nanosota-1* series bound potently to the SARS-CoV-2 RBD and were effective at inhibiting SARS-CoV-2 infection in vitro. The best performing drug candidate, *Nanosota-1C-Fc*, demonstrated preventative and therapeutic efficacy against SARS-CoV-2 infection in both hamster and mouse models. Produced at high yields, *Nanosota-1C-Fc* is easily scalable for mass production. It also demonstrated excellent in vitro thermostability, in vivo stability, and bioavailability. Our data suggest that *Nanosota-1c-Fc* can potentially contribute to the battle against COVID-19.

## Results

### *Nanosota-1* was identified by phage display

For the rapid identification of virus-targeting nanobodies, we constructed a naive nanobody phage display library using B cells isolated from the spleen, bone marrow, and blood of nearly a dozen non-immunized llamas and alpacas (*Figure 1*). Recombinant SARS-CoV-2 RBD, expressed and purified from mammalian cells, was screened against the library to identify RBD-targeting nanobodies. Selected nanobody clones were tested in a preliminary screen for their ability to neutralize SARS-CoV-2 pseudovirus entry into target cells (more details about the assay are reported below). The nanobody that demonstrated the highest preliminary neutralization potency was named *Nanosota-1A* and was subjected to two rounds of affinity maturation. For each round, random mutations were introduced to the whole gene of *Nanosota-1A* through error-prone polymerase chain reaction (PCR), and mutant phages were selected for enhanced binding to SARS-CoV-2 RBD. Nanobodies contain four framework regions (FRs) as structural scaffolds and three complementarity-determining regions (CDRs) for antigen binding. The nanobody after the first round of affinity maturation, *Nanosota-1B*, possessed one mutation in CDR3 and two other mutations in FR3 (near CDR3). Affinity maturation of *Nanosota-1B* resulted in *Nanosota-1C*, which possessed one mutation in CDR2 and another mutation in FR2. We next made an Fc-tagged version of *Nanosota-1C*, termed *Nanosota-1C-Fc*, to create a bivalent construct with increased molecular weight.

### *Nanosota-1* tightly bound to the SARS-CoV-2 RBD and completely blocked the binding of ACE2

To understand the structural basis for the binding of *Nanosota-1* to SARS-CoV-2 RBD, we determined the crystal structure of SARS-CoV-2 RBD complexed with *Nanosota-1C*. The structure showed that *Nanosota-1C* binds close to the center of the SARS-CoV-2 RBM (*Figure 2A*). Among the 14 RBM residues that directly interact with *Nanosota-1C*, six also directly interact with human ACE2 (*Figure 2—figure supplement 1*). When the structures of the RBD/*Nanosota-1C* complex and the RBD/ACE2 complex were superimposed together, significant clashes occurred between ACE2 and *Nanosota-1C* (*Figure 2B*), suggesting that *Nanosota-1C* binding to the RBD blocks ACE2 binding to the RBD. Moreover, trimeric SARS-CoV-2 spike protein is present in two different conformations: the RBD stands up in the open conformation but lies down in the closed conformation (*Shang et al., 2020b*; *Wrapp et al., 2020*; *Ke et al., 2020*). When the structures of the RBD/*Nanosota-1C* complex and the closed spike were superimposed, no clash was found between RBD-bound *Nanosota-1C* and the rest of the spike protein (*Figure 2—figure supplement 2A*). In contrast, severe clashes were identified between RBD-bound ACE2 and the rest of the spike protein in the closed conformation (*Figure 2—figure supplement 2B*). Additionally, neither RBD-bound *Nanosota-1C* nor RBD-bound ACE2 had clashes with the rest of the spike protein in the open conformation (*Figure 2—*

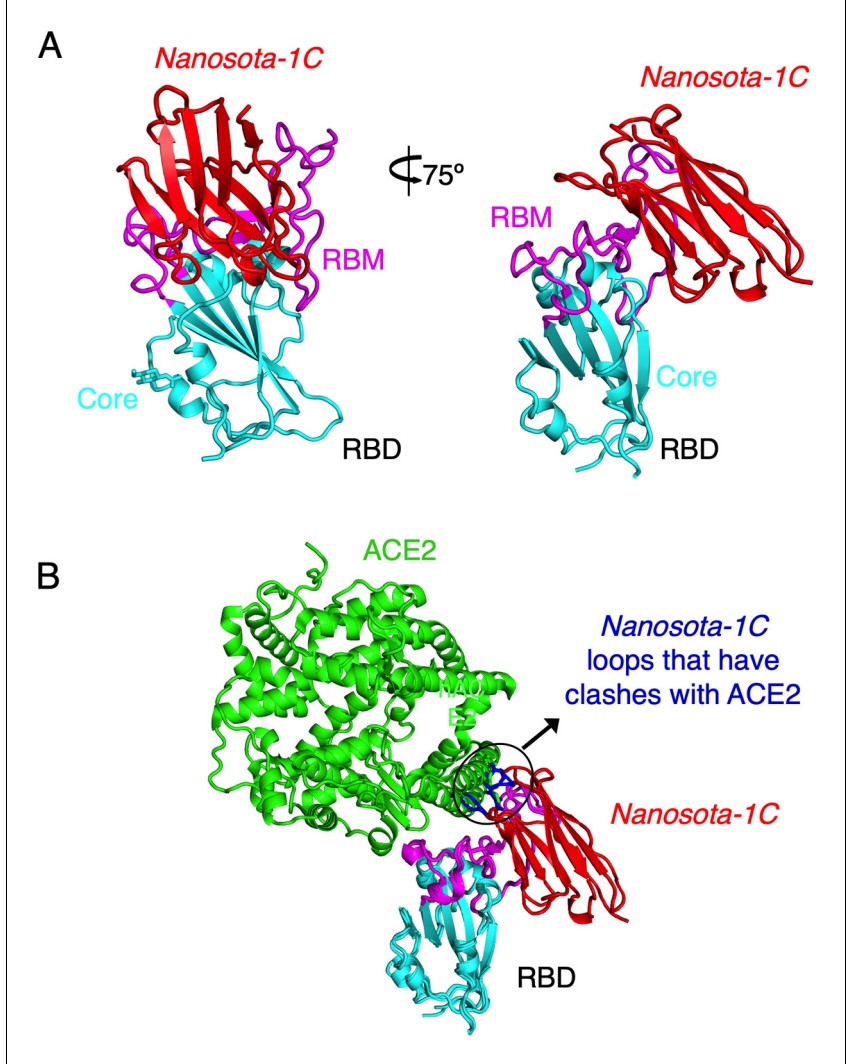

**Figure 2.** Crystal structure of SARS-CoV-2 RBD complexed with *Nanosota-1C*. (**A**) Structure of SARS-CoV-2 receptor-binding domain (RBD) complexed with *Nanosota-1C*, viewed at two different angles. *Nanosota-1C* is in red, the core structure of RBD is in cyan, and the receptor-binding motif (RBM) of RBD is in magenta. Structure data and refinement statistics are shown in *Table 2*. (**B**) Overlay of the structures of the RBD/*Nanosota-1C* complex and RBD/angiotensin-converting enzyme 2 (ACE2) complex (PDB 6M0J). ACE2 is in green. The structures of the two complexes were superimposed based on their common RBD structure. The *Nanosota-1C* loops that have clashes with ACE2 are in blue.

The online version of this article includes the following source data and figure supplement(s) for figure 2:

**Figure supplement 1.** Footprint of *Nanosota-1C* on SARS-CoV-2 RBD.

**Figure supplement 2.** The binding of *Nanosota-1C* to SARS-CoV-2 spike protein in different conformations.

**Figure supplement 3.** Measurement of the binding affinities between *Nanosota-1* and SARS-CoV-2 RBD by surface plasmon resonance assay using Biacore.

**Figure supplement 4.** Binding interactions between *Nanosota-1* and SARS-CoV-2 RBD.

**Figure supplement 4—source data 1.** Raw images for *Figure 2—figure supplement 4*.

---

*figure supplement 2C*; *Figure 2—figure supplement 2D*). Thus, *Nanosota-1C* can access the spike protein in both its open and closed conformations, whereas ACE2 can only access the spike protein in its open conformation. Overall, our structural data reveal that *Nanosota-1C* is an ideal RBD-targeting drug candidate that not only blocks virus binding to its receptor, but also accesses its target in the spike protein in different conformations.

To corroborate our structural data on RBD/*Nanosota-1* interactions, we performed binding experiments between *Nanosota-1* and SARS-CoV-2 RBD using recombinant ACE2 as a comparison. The binding affinity between *Nanosota-1* and the RBD was measured using surface plasmon resonance (*Table 1*; *Figure 2—figure supplement 3*). *Nanosota-1A, -1B, and -1C* bound to the RBD with increasing affinity ($K_d$ - 228–14 nM), confirming success of the stepwise affinity maturation. *Nanosota-1C-Fc* had the highest RBD-binding affinity ($K_d$ - 15.7 pM), which was ~3000 times tighter than the RBD-binding affinity of ACE2. Moreover, compared with ACE2, *Nanosota-1C-Fc* bound to the RBD with a higher $k_{on}$ and a lower $k_{off}$, demonstrating significantly faster binding and slower dissociation. Next, we investigated the competitive binding among *Nanosota-1C*, RBD, and ACE2 using protein pull-down assay (*Figure 2—figure supplement 4A*). *Nanosota-1C* and ACE2 were mixed together in different ratios in solution, with the concentration of ACE2 kept constant; RBD was added to pull down *Nanosota-1C* and ACE2 from solution. The result showed that as the concentration of *Nanosota-1C* increased, less ACE2 was pulled down by the RBD. Thus, *Nanosota-1C* and ACE2 bound competitively to the RBD. We repeated the above protein pull-down assay, with *Nanosota-1C-Fc* replacing *Nanosota-1C* (*Figure 2—figure supplement 4B*). The result confirmed that *Nanosota-1C-Fc* and ACE2 bound competitively to the RBD; it further showed that *Nanosota-1C-Fc* bound to the RBD much more strongly than ACE2 did, consistent with the binding affinity measurement. We then analyzed the competitive binding among *Nanosota-1C*, RBD, and ACE2 using gel filtration chromatography (*Figure 2—figure supplement 4C*). *Nanosota-1C*, RBD, and ACE2 were mixed, with both *Nanosota-1C* and ACE2 in molar excess over the RBD. Analysis by gel filtration chromatography documented that no ternary complex of *Nanosota-1C*, RBD, and ACE2 formed; instead, only binary complexes of RBD/ACE2 and RBD/*Nanosota-1C* were detected. Hence, the bindings of *Nanosota-1C* and ACE2 to the RBD are mutually exclusive.

## *Nanosota-1C-Fc* potently neutralized SARS-CoV-2 infection in vitro and in vivo

The ability of *Nanosota-1* to neutralize SARS-CoV-2 infection in vitro was investigated next. Both SARS-CoV-2 pseudovirus entry assay and live SARS-CoV-2 infection assay were performed (*Figure 3*). For the pseudovirus entry assay, retroviruses pseudotyped with SARS-CoV-2 spike protein (i.e., SARS-CoV-2 pseudoviruses) were used to enter human ACE2-expressing HEK293T cells in the presence of an inhibitor. The efficacy of the inhibitor was expressed as the concentration capable of neutralizing either 50 or 90% of the entry efficiency (i.e., $ND_{50}$ or $ND_{90}$, respectively). *Nanosota-1C-Fc* had an $ND_{50}$ of 0.27 µg/ml and an $ND_{90}$ of 3.12 µg/ml, both of which were ~10 times more potent than monovalent *Nanosota-1C* and the first of which was ~160 times more potent than ACE2 (*Figure 3A*). Additionally, *Nanosota-1* potently neutralized SARS-CoV-2 pseudovirus bearing the D614G mutation in the SARS-CoV-2 spike protein (*Figure 3—figure supplement 1*), which has become prevalent in many strains (*Korber et al., 2020*). For the live virus infection assay, live SARS-CoV-2 was used to infect Vero cells in the presence of an inhibitor. Efficacy of the inhibitor was described as the concentration capable of reducing the number of virus plaques by 50% (i.e., $ND_{50}$).

**Table 1.** Binding affinities between *Nanosota-1* and SARS-CoV-2 RBD as measured using surface plasmon resonance.
The previously determined binding affinity between human ACE2 and RBD is shown as a comparison (*Shang et al., 2020a*).

| | $K_d$ with SARS-CoV-2 RBD (M) | $k_{off}$ (s$^{-1}$) | $k_{on}$ (M$^{-1}$s$^{-1}$) |
|---|---|---|---|
| *Nanosota-1A* (before affinity maturation) | $2.28 \times 10^{-7}$ | $9.35 \times 10^{-3}$ | $4.10 \times 10^{4}$ |
| *Nanosota-1B* (after first round of affinity maturation) | $6.08 \times 10^{-8}$ | $7.19 \times 10^{-3}$ | $1.18 \times 10^{5}$ |
| *Nanosota-1C* (after second round of affinity maturation) | $1.42 \times 10^{-8}$ | $2.96 \times 10^{-3}$ | $2.09 \times 10^{5}$ |
| *Nanosota-1C-Fc* (after second round of affinity maturation; containing a C-terminal human Fc tag) | $1.57 \times 10^{-11}$ | $9.68 \times 10^{-5}$ | $6.15 \times 10^{6}$ |
| ACE2 | $4.42 \times 10^{-8}$ | $7.75 \times 10^{-3}$ | $1.75 \times 10^{5}$ |

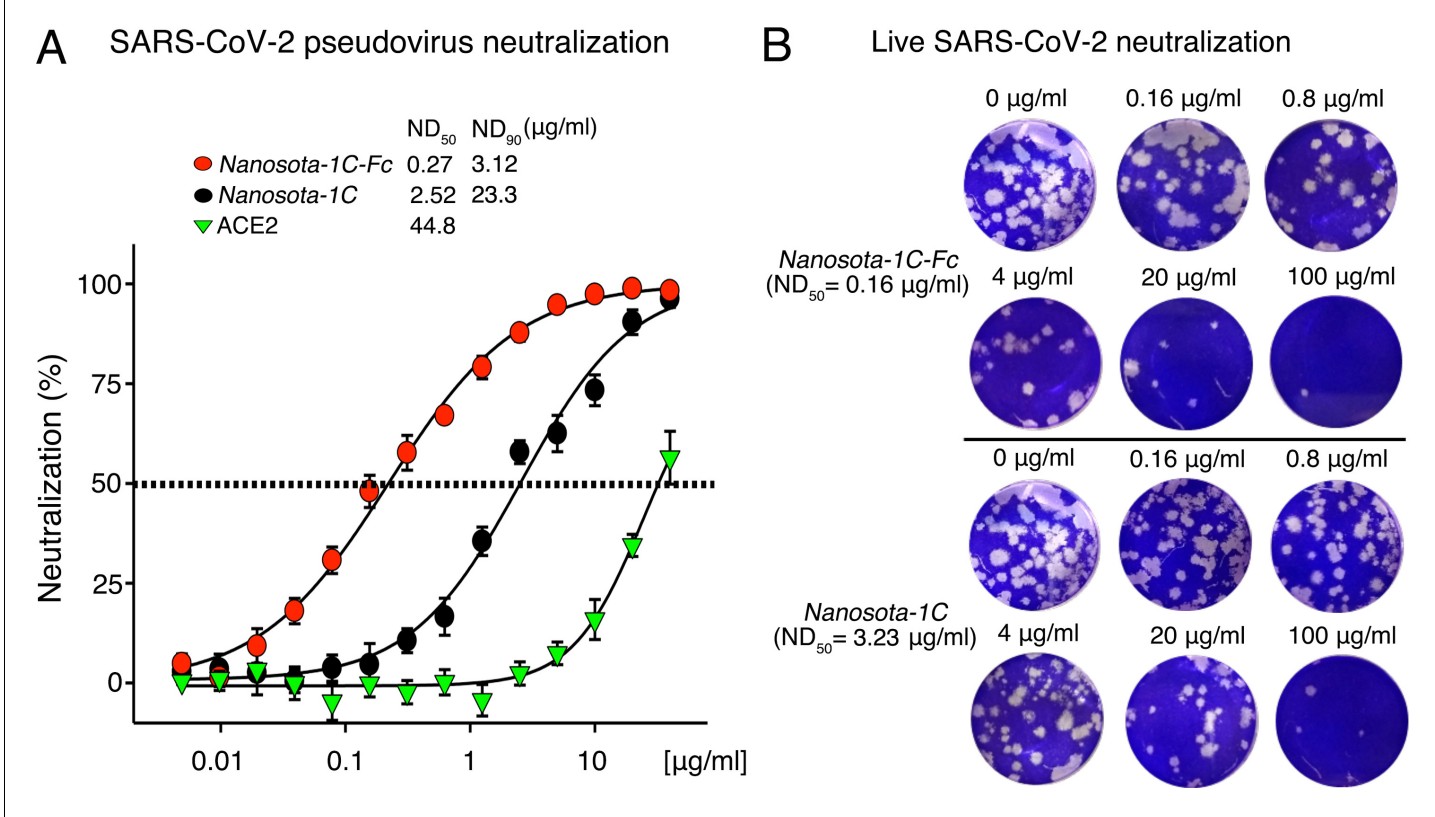

**Figure 3.** Efficacy of *Nanosota-1* in neutralizing SARS-CoV-2 infections in vitro. (**A**) Neutralization of SARS-CoV-2 pseudovirus entry into target cells by one of three inhibitors: *Nanosota-1C-Fc, Nanosota-1C*, and recombinant human angiotensin-converting enzyme 2 (ACE2). Retroviruses pseudotyped with SARS-CoV-2 spike protein (i.e., SARS-CoV-2 pseudoviruses) were used to enter HEK293T cells expressing human ACE2 in the presence of individual inhibitors at various concentrations. Entry efficiency was characterized via a luciferase signal, indicating successful cell entry. Data are the mean ± SEM (n = 4). Nonlinear regression was performed using a log (inhibitor) versus normalized response curve and a variable slope model ($R^2 > 0.95$ for all curves). The efficacy of each inhibitor was expressed as the concentration capable of neutralizing pseudovirus entry by 50% (i.e., $ND_{50}$) or 90% (i. e., $ND_{90}$). $ND_{90}$ for ACE2 was not calculated due to insufficient data points. The assay was repeated three times (biological replication: new aliquots of pseudoviruses and cells were used for each repeat). (**B**) Neutralization of live SARS-CoV-2 infection of target cells by one of two inhibitors: *Nanosota-1C-Fc* and *Nanosota-1C*. The potency of *Nanosota-1* in neutralizing live SARS-CoV-2 infections was evaluated using a SARS-CoV-2 plaque-reduction neutralization test (PRNT) assay. 80 plaque-forming unit (PFU) infectious SARS-CoV-2 particles were used to infect Vero E6 cells in the presence of individual inhibitors at various concentrations. Infection was characterized as the number of virus plaques formed in overlaid cells. Images of virus plaques for each inhibitor at the indicated concentrations are shown. Each image represents data from triplications. The efficacy of each inhibitor was calculated and expressed as the concentration capable of reducing the number of virus plaques by 50% (i.e., $ND_{50}$). The assay was repeated twice (biological replication: new aliquots of virus particles and cells were used for each repeat).

The online version of this article includes the following figure supplement(s) for figure 3:

**Figure supplement 1.** Neutralization of SARS-CoV-2 pseudovirus, which contains the D614G mutation in the spike protein, by *Nanosota-1*.

**Figure supplement 2.** Detailed data on the neutralization of live SARS-CoV-2 infection of target cells by *Nanosota-1*.

*Nanosota-1C-Fc* had an $ND_{50}$ of 0.16 μg/ml, which was significantly more potent than monovalent *Nanosota-1C* and ACE2 (***Figure 3B***; ***Figure 3—figure supplement 2***). Overall, both *Nanosota-1C-Fc* and *Nanosota-1C* are potent inhibitors of SARS-CoV-2 pseudovirus entry and live SARS-CoV-2 infection.

After the in vitro studies, we next evaluated the therapeutic efficacy of the lead drug candidate *Nanosota-1C-Fc* in a hamster model challenged with SARS-CoV-2 (at a titer of 1 x 10⁶ Median Tissue Culture Infectious Dose [$TCID_{50}$]) via intranasal inoculation. Three experimental groups of hamsters (six per group) received a single dose of *Nanosota-1C-Fc* via intraperitoneal injection: (i) 24-hr pre-challenge at 20 mg/kg body weight, (ii) 4-hr post-challenge at 20 mg/kg, and (iii) 4-hr post-challenge at 10 mg/kg. As previously validated in this model (***Sia et al., 2020***), body weight and tissue pathology were used as metrics of therapeutic efficacy. In an untreated control group that received

phosphate-buffered saline (PBS), weight loss precipitously started on day 1 post-challenge and the lowest weights were recorded on day 6 (*Figure 4A*). Pathology analysis on tissues collected on day 10 revealed moderate hyperplasia in the bronchial tubes (i.e., bronchioloalveolar hyperplasia) (*Figure 4B*), with little hyperplasia in the lungs. These data are consistent with previous reports showing that SARS-CoV-2 mainly infects the bronchial epithelial cells of this hamster model (*Sia et al., 2020*). In contrast, hamsters that received *Nanosota-1C-Fc* 24 hr pre-challenge were protected from SARS-CoV-2, as evidenced by no weight loss and no bronchioloalveolar hyperplasia (*Figure 4A*; *Figure 4B*). When administered 4 hr post-challenge, *Nanosota-1C-Fc* also effectively protected hamsters from SARS-CoV-2 infections at either dosage (20 or 10 mg/kg), as evidenced by the favorable therapeutic metrics (*Figure 4A*; *Figure 4B*). Overall, *Nanosota-1C-Fc* was effective at curtailing SARS-CoV-2 infections preventively and therapeutically in the hamster model.

To further examine the in vivo efficacy of *Nanosota-1C-Fc*, we evaluated its therapeutic efficacy in human ACE2-transgenic mice challenged with SARS-CoV-2 (at a titer of $5 \times 10^3$ plaque-forming unit [PFU]) via intranasal inoculation. Instead of monitoring the body weights of the mice through the viral infection and recovery process, we measured the virus titers in the lungs at the peak of the viral infection. To this end, four experimental groups of mice (seven per group) received a single dose of *Nanosota-1C-Fc* via intraperitoneal injection: (i) 24-hr pre-challenge at 20 mg/kg body weight, (ii) 24-hr pre-challenge at 10 mg/kg body weight, (iii) 4-hr post-challenge at 20 mg/kg, and (iv) 4-hr post-challenge at 10 mg/kg. Five out of the seven mice from each group were euthanized on day 2 post-challenge, and the virus titers in their lungs were measured using a virus titer plaque assay (*Figure 4C*). Compared to the untreated control group, the mice that received *Nanosota-1C-Fc* had much lower virus titers in the lungs (~1000 times lower in the pre-challenge groups and ~100 times lower in the post-challenge groups). In addition to the above virus titer measurements, the remaining two mice in each group were euthanized on day 5 post-challenge for pathologic analysis of lung tissues (*Figure 4D*). In the untreated control group, histological examination revealed extensive inflammatory cell infiltration, especially in the peribronchial region, alveolar edema, and proliferative alveolar epithelium. These results are consistent with previous reports on the SARS-CoV-2-induced lung pathology of these mice (*Zheng et al., 2021*). The mice that received *Nanosota-1C-Fc* showed a near absence of lung pathology for both the pre- and post-challenge groups at 20 mg/kg body weight and minor lung pathological changes for the pre-challenge group at 10 mg/kg body weight. Pathological changes were still present in mice treated post-challenge with 10 mg/kg *Nanosota-1C-Fc*. Overall, *Nanosota-1C-Fc* effectively protected the mouse model from SARS-CoV-2 infection of their lungs.

## *Nanosota-1C-Fc* is stable in vitro and in vivo with excellent bioavailability

With the lead drug candidate *Nanosota-1C-Fc* demonstrating therapeutic efficacy in vivo, we characterized other parameters important for its clinical translation. First, we expressed *Nanosota-1C-Fc* in bacteria (*Figure 5A*). After purification on protein A column and gel filtration, the purity of *Nanosota-1C-Fc* was nearly 100%. With no optimization, the expression yield reached 40 mg/l of bacterial culture. Second, we investigated the in vitro stability of *Nanosota-1C-Fc* incubated at four temperatures (−80°C, 4°C, 25°C, or 37°C) for 1 week and then measured its remaining SARS-CoV-2 RBD-binding capacity using enzyme-linked immunosorbent assay (ELISA) (*Figure 5B*). With −80°C as a baseline, *Nanosota-1C-Fc* retained nearly all of its RBD-binding capacity at the temperatures surveyed. Third, we measured the in vivo stability of *Nanosota-1C-Fc* (*Figure 5C*). *Nanosota-1C-Fc* was injected into mice via tail vein. Sera were obtained at different time points and measured for their SARS-CoV-2 RBD-binding capacity using ELISA. *Nanosota-1C-Fc* retained significant RBD-binding capability after 10 days in vivo. In contrast, *Nanosota-1C* was stable for only several hours in vivo (*Figure 5—figure supplement 1A*; *Figure 5—figure supplement 1B*). Last, we examined the biodistribution of *Nanosota-1C-Fc* in mice (*Figure 5D*). *Nanosota-1C-Fc* was radiolabeled with zirconium-89 ($^{89}$Zr) and injected systemically into mice. Tissues were collected at various time points and biodistribution of *Nanosota-1C-Fc* was quantified using a scintillation counter. After 3 days, *Nanosota-1C-Fc* remained at high levels in the blood, lung, heart, kidney, liver, and spleen, all of which are targets for SARS-CoV-2 (*Puelles et al., 2020*); moreover, it remained at low levels in the intestine, muscle, and bones. In contrast, *Nanosota-1C* had poor biodistribution, documenting high renal

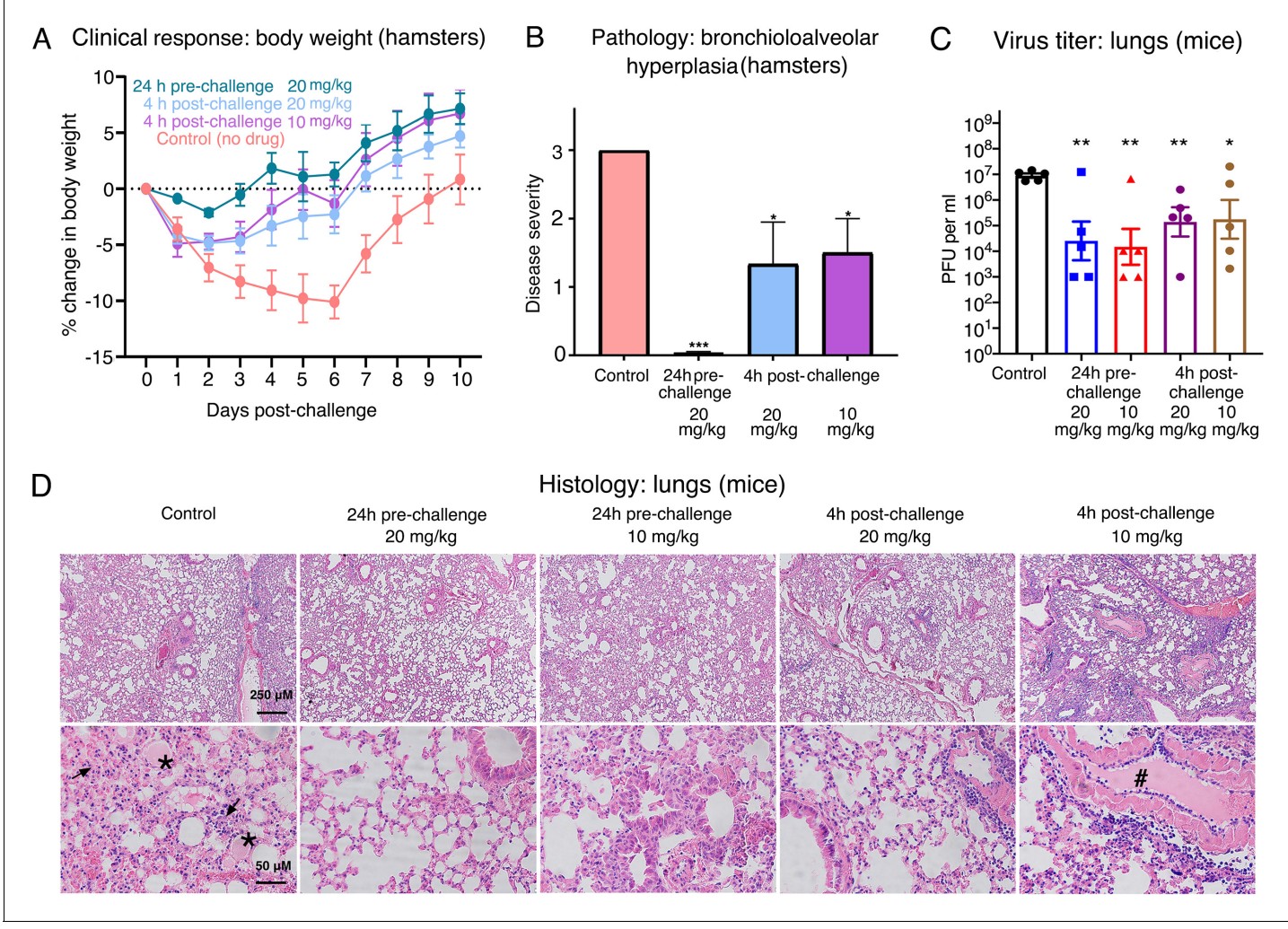

**Figure 4.** Efficacy of *Nanosota-1* in protecting both hamsters and mice from SARS-CoV-2 infections. (**A**, **B**) Hamsters (six per group) were injected with a single dose of *Nanosota-1C-Fc* at the indicated time point and the indicated dosage. At day 0, all groups (experimental and control) were challenged with SARS-CoV-2 (at a titer of $10^6$ Median Tissue Culture Infectious Dose [TCID$_{50}$]). (**A**) Body weights of hamsters were monitored on each day and percent change in body weight relative to day 0 was calculated for each hamster. Data are the mean ± SEM (n = 6). Analysis of variance (ANOVA) on group as a between-group factor and day (1-10) as a within-group factor revealed significant differences between the control group and each of the following groups: 24-hr pre-challenge (20 mg/kg) group ($F$(1, 10) = 17.80, p = 0.002; effect size $\eta_p^2$ = 0.64), 4-hr post-challenge (20 mg/kg) group ($F$(1, 10) = 5.02, p = 0.035; $\eta_p^2$ = 0.37), and 4-hr post-challenge (10 mg/kg) group ($F$(1, 10) = 7.04, p = 0.024; $\eta_p^2$ = 0.41). All p-values are two-tailed. (**B**) Tissues of bronchial tubes from each of the hamsters were collected on day 10 and scored for the severity of bronchioloalveolar hyperplasia: 3 - moderate; 2 - mild; 1 - minimum; 0 - none. Data are the mean ± SEM (n = 6). A comparison between the control group and each of other groups was performed using one-tailed Student's t-test for directional tests. ***p<0.001; *p<0.05. (**C**, **D**) Human ACE2-transgenic mice (seven per group) were injected with a single dose of *Nanosota-1C-Fc* at the indicated time point and the indicated dosage. At day 0, all groups (experimental and control) were challenged with SARS-CoV-2 (at a titer of 5 x $10^3$ plaque-forming unit [PFU]). (**C**) Five mice from each group were euthanized on day 2 post-challenge, and the virus titers in their lungs were measured using a plaque assay. A comparison between the control group and each of the other groups was performed using one-tailed Student's t-test for directional tests. Data are the mean ± SEM (n = 5). **p<0.01; *p<0.05. (**D**) The remaining two mice from each group were euthanized on day 5 post-challenge. Lung tissues were collected and examined for pathological changes after staining with hematoxylin and eosin. Phosphate-buffered saline (PBS) control group: severe inflammatory cell infiltration and bronchiole infiltration on the top panel; severe alveolar edema filled with liquid (labeled *) and proliferative alveolar epithelium (labeled →) on the bottom panel. 24-hr pre-challenge 20 mg/kg group: close to normal. 24-hr pre-challenge 10 mg/kg group: minor proliferative alveolar epithelium and inflammatory cell infiltration. 4-hr post-challenge 20 mg/kg group: close to normal. 4-hr post-challenge 10 mg/kg group: obvious cell inflammatory cell infiltration and vascular thrombosis (labeled #).

The online version of this article includes the following source data for figure 4:

**Source data 1.** Raw images for *Figure 4*.

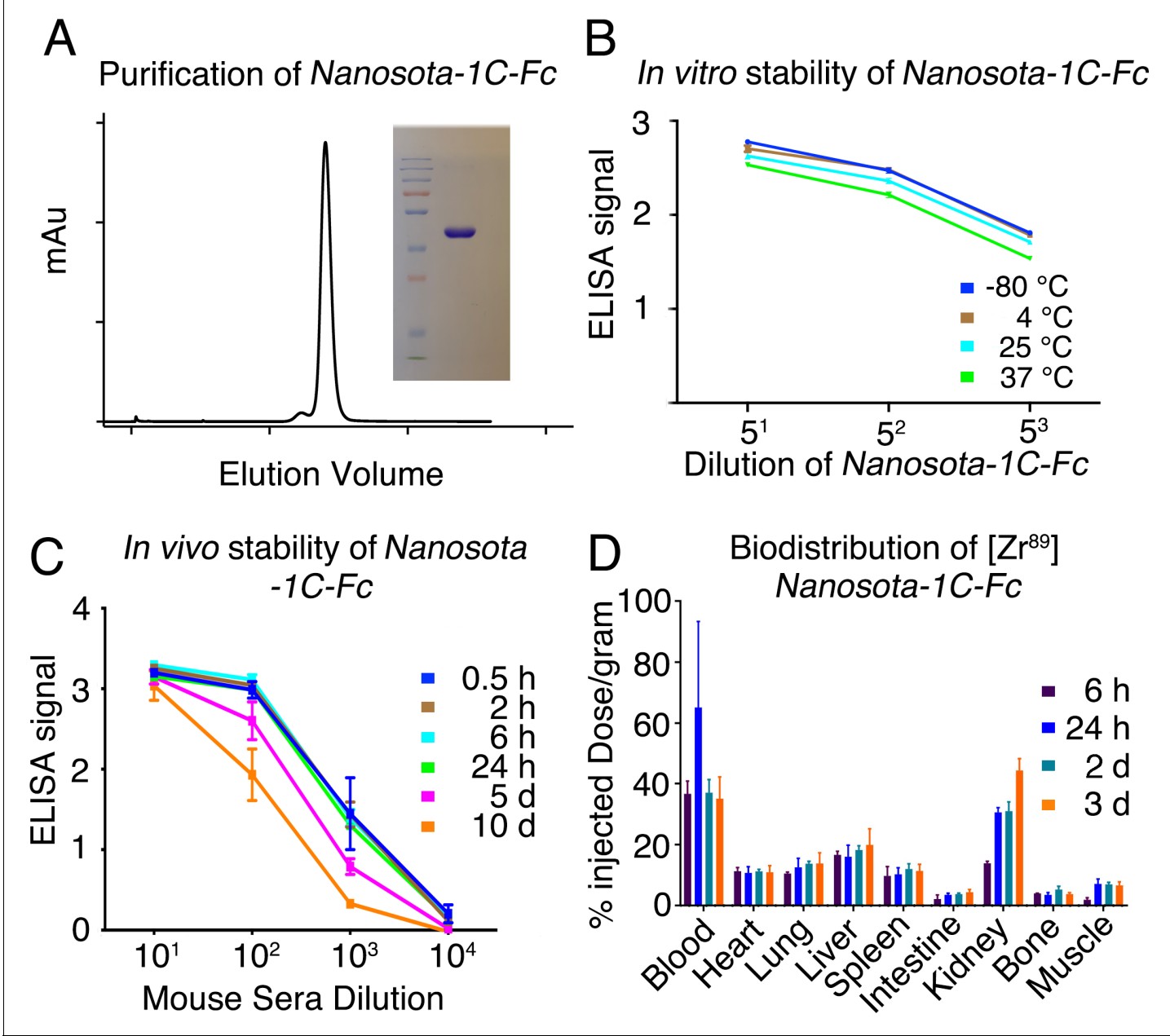

**Figure 5.** Analysis of expression, purification, and pharmacokinetics of *Nanosota-1C-Fc*. (**A**) Purification of *Nanosota-1C-Fc* from bacteria. The protein was nearly 100% pure after gel filtration chromatography, as demonstrated by its elution profile and sodium dodecyl sulfate-polyacrylamide gel electrophoresis (SDS-PAGE) (stained by Coomassie blue). The yield of the protein was 40 mg/l of bacterial culture, without any optimization of the expression. (**B**) In vitro stability of *Nanosota-1C-Fc*. The protein was stored at indicated temperatures for a week, and then a dilution enzyme-linked immunosorbent assay (ELISA) was performed to evaluate its SARS-CoV-2 receptor-binding domain (RBD)-binding capability. Data are the mean ± SEM (n = 4). (**C**) In vivo stability of *Nanosota-1C-Fc*. *Nanosota-1C-Fc* was injected into mice, mouse sera were collected at various time points, and *Nanosota-1C-Fc* remaining in the sera was detected for its SARS-CoV-2 RBD-binding capability as displayed in a dilution ELISA. Data are the mean ± SEM (n = 3). (**D**) Biodistribution of [89Zr]Zr-*Nanosota-1C-Fc*. *Nanosota-1C-Fc* was radioactively labeled with zirconium-89 (89Zr) and injected into mice via the tail vein. Different tissues or organs were collected at various time points (n = 3 mice per time point). The amount of *Nanosota-1C-Fc* present in each tissue or organ was measured through examining the radioactive count of each tissue or organ. Data are the mean ± SEM (n = 3).

The online version of this article includes the following source data and figure supplement(s) for figure 5:

**Source data 1.** Raw images for *Figure 5*.

**Figure supplement 1.** Pharmacokinetics of *Nanosota-1C*.

clearance (*Figure 5—figure supplement 1C*). Overall, our findings suggest that *Nanosota-1C-Fc* is a potent anti-SARS-CoV-2 drug candidate with translational values.

## Discussion

Nanobody therapeutics derived from camelid antibodies has advantages relative to conventional antibodies. Although several studies have reported nanobody drug candidates that specifically target SARS-CoV-2 (*Huo et al., 2020*; *Hanke et al., 2020*; *Xiang et al., 2020*; *Schoof et al., 2020*; *Wrapp et al., 2020a*; *Pymm et al., 2021*; *Nambulli et al., 2021*), as of this writing, only two studies have evaluated their nanobody drug candidates in an animal model for anti-SARS-CoV-2 therapeutic efficacy (*Pymm et al., 2021*; *Nambulli et al., 2021*). None of these studies have evaluated their nanobody drug candidates for in vitro thermostability, in vivo stability, or tissue biodistribution. The current study describes the full development and characterization of a series of nanobody drug candidates that specifically target the SARS-CoV-2 RBD. Our study included screening of nanobody phage display library, two rounds of affinity maturation, structural determination of the RBD/*Nanosota-1* complex, and neutralizations of SARS-CoV-2 pseudovirus entry and live SARS-CoV-2 infection in vitro. We also evaluated the protection efficacy in two different animal models (hamsters and mice), characterized the nanobody's production yields, demonstrated its in vitro thermostability and in vivo stability, and clarified its tissue biodistribution. The extensive scope of the work makes the current study among the most comprehensive on anti-SARS-CoV-2 nanobody drug candidates.

The two best-performing drug candidates from our study are *Nanosota-1C* and *Nanosota-1C-Fc*. The latter was constructed from the former through addition of an Fc tag to make a bivalent molecule with increased molecular weight and picomolar RBD-binding affinity. In this study, recombinant viral receptor ACE2 was selected as a comparison for evaluating antiviral potency because *Nanosota-1* directly competes with ACE2 for the same binding site on the RBD. Compared with ACE2, *Nanosota-1C-Fc* bound to the RBD ~3000-fold more strongly and inhibited SARS-CoV-2 pseudovirus entry ~160-fold more effectively. Compared with ACE2, the much higher anti-SARS-CoV-2 potency of *Nanosota-1C-Fc* was partly due to the small size of its antigen-binding domain and its ideal binding site on the RBD, allowing *Nanosota-1C-Fc* to access the RBD in both the open spike during viral infection and the closed spike during viral immune evasion. Thus, the *Nanosota-1* series are ideal RBD-targeting drug candidates that can inhibit SARS-CoV-2 viral particles regardless of whether the viral spike molecules are in an open or a closed conformation. Importantly, we showed that the effectiveness of *Nanosota-1C-Fc* was not limited to in vitro experiments, but translated directly to in vivo experiments by demonstrating efficacy in animal models. It is worth noting that the molecular weight of *Nanosota-1C-Fc* (78 kDa) is above the kidney clearance threshold (60 kDa) (*Steeland et al., 2016*), but it's still only half of conventional antibodies' molecular weight (150 kDa). Its ideal size contributed to ease of production, excellent in vitro thermostability, good in vivo stability, and high tissue bioavailability. All of these features are critical for the implementation of *Nanosota-1C-Fc* as a potential COVID-19 therapeutic.

How can *Nanosota-1* drug candidates contribute to the battle against COVID-19? First, as evidenced by both of our animal studies, *Nanosota-1C-Fc* can be used to prevent SARS-CoV-2 infections. Because of its excellent in vivo stability (significant RBD-binding capacity remained after 10 days in vivo), a single injected dose of *Nanosota-1C-Fc* can theoretically protect a person from SARS-CoV-2 infection for days or weeks in the outpatient setting, reducing the spread of SARS-CoV-2 in human populations. Second, data from both of our animal studies showed that *Nanosota-1C-Fc* can be used to treat SARS-CoV-2 infections, thus, potentially saving lives and alleviating symptoms in infected patients in the clinical setting. It is worth noting that given the recent success of COVID-19 vaccines, future development of *Nanosota-1* series should focus on its therapeutic efficacy rather than its preventive efficacy. Moreover, to combat emerging SARS-CoV-2 variants with RBM mutations, *Nanosota-1* series can be further developed through additional affinity maturation against the RBD of SARS-CoV-2 variants. The series may also be developed to become part of cocktail therapies that target multiple sites on SARS-CoV-2. Third, despite its rapid clearance from the blood, *Nanosota-1C* could be used as an inhaler to treat infections in the respiratory tracts (*Van Heeke et al., 2017*; *Nambulli et al., 2021*) or as an oral drug to treat infections in the intestines (*Vega et al., 2013*). The worldwide distribution of COVID-19 in the world calls for the large-scale manufacturing

of anti-SARS-CoV-2 therapeutics. Molecules such as *Nanosota-1*, with their high production yields from bacteria and excellent in vitro and in vivo stabilities, are promising drug candidates to meet this need. Therefore, if further validated in clinical trials, *Nanosota-1* drug candidates can help minimize the mortality and morbidity of SARS-CoV-2 infections and contribute to the battle against COVID-19.

## Materials and methods

### Cell lines, plasmids, and virus

HEK293T cells (American Type Culture Collection [ATCC]) were cultured in Dulbecco's modified Eagle medium (DMEM) supplemented with 10% fetal bovine serum (FBS), 2 mM L-glutamine, 100 units/ml penicillin, and 100 μg/ml streptomycin (Life Technologies). Vero E6 cells (ATCC) were grown in Eagle's minimal essential medium (EMEM) supplemented with penicillin (100 units/ml), streptomycin (100 μg/ml), and 10% FBS. ss320 *Escherichia* coli (Lucigen), TG1 *E. coli* (Lucigen), and SHuffle T7 *E. coli* (New England Biolabs) were grown in TB medium or 2YT medium with 100 mg/l ampicillin. HEK293T cells were authenticated by ATCC using short tandem repeat (STR) profiling. HEK293T cells and Vero E6 cells tested negative for mycoplasma contamination. No commonly misidentified cell lines were used.

SARS-CoV-2 spike (GenBank accession number QHD43416.1) and ACE2 (GenBank accession number NM_021804) were described previously (*Shang et al., 2020a*). SARS-CoV-2 RBD (residues 319–529) was subcloned into Lenti-CMV vector (Vigene Biosciences) with an N-terminal tissue plasminogen activator (tPA) signal peptide and a C-terminal human IgG4 Fc tag or His tag. The ACE2 ectodomain (residues 1–615) was constructed in the same way except that its own signal peptide was used. *Nanosota-1A*, *-1B*, and *-1C* were each cloned into PADL22c vector (Lucigen) with an N-terminal PelB leader sequence and C-terminal His tag and HA tag. *Nanosota-1C-Fc* was cloned into pET42b vector (Novagen) with a C-terminal human IgG$_1$ Fc tag.

SARS-CoV-2 (US_WA-1 isolate) from CDC (Atlanta) was used throughout the study. Experiments involving infectious SARS-CoV-2 were conducted at the University of Texas Medical Branch and the University of Iowa in approved biosafety level 3 laboratories.

### Construction of camelid nanobody phage display library

The camelid nanobody phage display library was constructed as previously described (*Abbady et al., 2011*; *Olichon and de Marco, 2012*). Briefly, total mRNA was isolated from B cells from the spleen, bone marrow, and blood of over a dozen non-immunized llamas and alpacas. cDNA was prepared from the mRNA. The cDNA was then used in nested PCRs to construct the DNA for the library. The first PCR was to amplify the gene fragments encoding the variable domain of the nanobody. The second PCR (PCR2) was used to add restriction sites (SFI-I), a PelB leader sequence, a His$_6$ tag, and an HA tag. The PCR2 product was digested with SFI-I (New England Biolabs) and then ligated with SFI-I-digested PADL22c vector. The ligated product was transformed via electroporation into TG1 *E. coli* (Lucigen). Aliquots of cells were spread onto 2YT agar plates supplemented with ampicillin and glucose, incubated at 30°C overnight, and then scraped into 2YT media. After centrifugation, the cell pellet was suspended into 50% glycerol and stored at −80°C. The library size was $7.5 \times 10^{10}$. To display nanobodies on phages, aliquots of the TG1 *E. coli* bank were inoculated into 2YT media, grown to early logarithmic phase, and infected with M13K07 helper phage.

### Camelid nanobody library screening

The above camelid nanobody phage display library was used in the bio-panning as previously described (*Hintz et al., 2019*). Briefly, four rounds of panning were performed to obtain the SARS-CoV-2 RBD-targeting nanobodies with high RBD-binding affinity. The amounts of the RBD antigen used in coating the immune tubes in each round were 75 μg, 50 μg, 25 μg, and 10 μg, respectively. The retained phages were eluted using 1 ml 100 mM triethylamine and neutralized with 500 μl 1 M Tris-HCl, pH 7.5. The eluted phages were amplified in TG1 *E. coli* and rescued with M13K07 helper phage. The eluted phages from round 4 were used to infect ss320 *E. coli*. Single colonies were picked into 2YT media and nanobody expressions were induced with 1 mM isopropyl-beta-D-

thiogalactopyranoside (IPTG). The supernatants were subjected to ELISA for selection of strong binders (described below). The strong binders were then expressed and purified (described below) and subjected to SARS-CoV-2 pseudovirus entry assay for selection of anti-SARS-CoV-2 efficacy (described below). The lead nanobody after initial screening was named *Nanosota-1A*.

## Affinity maturation

Affinity maturation of *Nanosota-1A* was performed as previously described (*Hust and Lim, 2018*). Briefly, mutations were introduced into the whole gene of *Nanosota-1A* using error-prone PCR. Two rounds of error-prone PCR were performed using the GeneMorph II Random Mutagenesis Kit (Agilent Technologies). The PCR product was cloned into the PADL22c vector and transformed via electroporation into the TG1 *E. coli*. The library size was 6 x $10^8$. Three rounds of bio-panning were performed using 25 ng, 10 ng, and 2 ng RBD-Fc, respectively. The strongest binder after affinity maturation was named *Nanosota-1B*. A second round of affinity maturation was performed in the same way as the first round, except that three rounds of bio-panning were performed using 10 ng, 2 ng, and 0.5 ng RBD-Fc, respectively. The strongest binder after the second round of affinity maturation was named *Nanosota-1C*.

## Production of *Nanosota-1*

*Nanosota-1A*, *-1B*, and *-1C* were each purified from the periplasm of ss320 *E. coli* after the cells were induced by 1 mM IPTG. The cells were collected and re-suspended in 15 ml TES buffer (0.2 M Tris, pH 8, 0.5 mM ethylenediaminetetraacetic acid (EDTA), 0.5 M sucrose), shaken on ice for 1 hr, and then incubated with 40 ml TES buffer (diluted 4 times) followed by shaking on ice for another 1 hr. The protein in the supernatant was sequentially purified using a Ni-NTA column and a Superdex200 gel filtration column (GE Healthcare) as previously described (*Shang et al., 2020a*). *Nanosota-1C-Fc* was purified from the cytoplasm of Shuffle T7 *E. coli*. The induction of protein expression was the same as above. After induction, the cells were collected, re-suspended in PBS, and disrupted using Branson Digital Sonifier (Thermofisher). The protein in the supernatant was sequentially purified on protein A column and Superdex200 gel filtration column as previously described (*Shang et al., 2020a*).

## Production of SARS-CoV-2 RBD and ACE2

HEK293T cells stably expressing SARS-CoV-2 RBD (containing a C-terminal His tag or Fc tag) or human ACE2 ectodomain (containing a C-terminal His tag) were made according to the E and F sections of the pLKO.1 Protocol from Addgene (http://www.addgene.org/protocols/plko/). The proteins were secreted to cell culture media, harvested, and purified on either a Ni-NTA column (for His-tagged proteins) or a protein A column (for Fc-tagged proteins) and then on a Superdex200 gel filtration column as previously described (*Shang et al., 2020a*).

## ELISA

ELISA was performed to detect the binding between SARS-CoV-2 RBD and *Nanosota-1* (either as purified recombinant proteins or as proteins in the mouse serum) as previously described (*Zhao et al., 2018*). Briefly, ELISA plates were coated with recombinant SARS-CoV-2 RBD-His or RBD-Fc and were then incubated sequentially with nanobody proteins, horseradish peroxidase (HRP)-conjugated anti-llama antibody (1:5000) (Sigma), or HRP-conjugated anti-human-Fc antibody (1:5000) (Jackson ImmunoResearch). ELISA substrate (Invitrogen) was added to the plates, and the reactions were stopped with 1N $H_2SO_4$. The absorbance at 450 nm ($A_{450}$) was measured using a Synergy LX Multi-Mode Reader (BioTek).

## Determination of the structure of SARS-CoV-2 RBD complexed with *Nanosota-1C*

To prepare the RBD/*Nanosota-1C* complex for crystallization, the two proteins were mixed together in solution and purified using a Superdex200 gel filtration column (GE Healthcare). The complex was concentrated to 10 mg/ml in buffer—20 mM Tris, pH 7.2, and 200 mM NaCl. Crystals were screened at High-Throughput Crystallization Screening Center (Hauptman-Woodward Medical Research Institute) as previously described (*Luft et al., 2003*) and were grown in sitting drops at room

temperature over wells containing 50 mM $MnCl_2$, 50 mM MES, pH 6.0, and 20% (W/V) polyethylene glycol (PEG) 4000. Crystals were soaked briefly in 50 mM $MnCl_2$, 50 mM MES, pH 6.0, 25% (W/V) PEG 4000%, and 30% ethylene glycol before being flash-frozen in liquid nitrogen. X-ray diffraction data were collected at the Advanced Photon Source beamline 24-ID-E. The structure was determined by molecular replacement using the structures of SARS-CoV-2 RBD (PDB 6M0J) and another nanobody (PDB 6QX4) as the search templates. Structure data and refinement statistics are shown in *Table 2*.

## Surface plasmon resonance assay

Surface plasmon resonance assay using a Biacore S200 system (GE Healthcare) was carried out as previously described (*Shang et al., 2020a*). Briefly, SARS2-CoV-2 RBD-His was immobilized to a CM5 sensor chip (GE Healthcare). Serial dilutions of purified recombinant *Nanosota-1* proteins were injected at different concentrations: 320–10 nM for *Nanosota-1A*; 80–2.5 nM for *Nanosota-1B* and *Nanosota-1C*; 20–1.25 nM for *Nanosota-1C-Fc*. The resulting data were fit to a 1:1 binding model using Biacore Evaluation Software (GE Healthcare).

## Protein pull-down assay

Protein pull-down assay was performed using an immunoprecipitation kit (Invitrogen) as previously described (*Shang et al., 2020a*). Briefly, to pull down ACE2-His (containing a C-terminal His tag) and *Nanosota-1C* (containing a C-terminal His tag), 10 μl protein A beads were incubated with 1 μg SARS-CoV-2 RBD-Fc at room temperature for 1 hr. Then different amounts (7.04, 3.52. 1.76, 0.88, 0.44, 0.22, or 0 μg) of *Nanosota-1C* and 4 μg ACE2-His were added to the RBD-bound beads. After incubation at room temperature for 1 hr, the bound proteins were eluted using elution buffer (0.1 M glycine, pH 2.7). The samples were then subjected to sodium dodecyl sulfate-polyacrylamide gel electrophoresis (SDS-PAGE) and analyzed through western blot using an anti-His antibody.

To pull down ACE2-His and *Nanosota-1C-Fc* (containing a C-terminal Fc tag), 10 μl streptavidin beads were incubated with 1 μg SARS-CoV-2 RBD-His (biotinylated using EZ-LinkTM Sulfo-NHS-LC-Biotinylation Kit; Thermo Scientific) at room temperature for 1 hr. Then different amounts (17.80, 8.90. 4.45, 2.23, 1.11, 0.56, or 0 μg) of *Nanosota-1C-Fc* and 4 μg ACE2-His were added to the RBD-bound beads. After incubation at room temperature for 1 hr, the bound proteins were eluted using elution buffer (0.1 M glycine, pH 2.7). The samples were then subjected to SDS-PAGE and analyzed through western blot using an anti-His antibody (for detecting ACE2-His) or an anti-Fc antibody (for detecting *Nanosota-1C-Fc*).

## Gel filtration chromatography assay

Gel filtration chromatography assay was performed on a Superdex200 column. 500 μg human ACE2, 109 μg *Nanosota-1C*, and 121 μg SARS-CoV-2 RBD were incubated together at room temperature for 30 min. The mixture was subjected to gel filtration chromatography. Samples from each peak off the column were then subjected to SDS-PAGE and analyzed through Coomassie blue staining.

## SARS-CoV-2 pseudovirus entry assay

The potency of *Nanosota-1* in neutralizing SARS-CoV-2 pseudovirus entry was evaluated as previously described (*Shang et al., 2020a*; *Shang et al., 2020b*). Briefly, HEK293T cells were co-transfected with a plasmid carrying an Env-defective, luciferase-expressing human immunodeficiency virus-1 (HIV-1) genome (pNL4-3.luc.R-E-) and pcDNA3.1(+) plasmid encoding SARS-CoV-2 spike protein. Pseudoviruses were collected 72 hr after transfection, incubated with individual inhibitors at different concentrations at 37°C for 1 hr, and then used to enter HEK293T cells expressing human ACE2. After pseudoviruses and target cells were incubated together at 37°C for 6 hr, the medium was changed to fresh medium, followed by incubation for another 60 hr. Cells were then washed with PBS buffer and lysed. Aliquots of cell lysates were transferred to plates, followed by the addition of luciferase substrate. Relative light units (RLUs) were measured using an EnSpire plate reader (PerkinElmer). The efficacy of each inhibitor was expressed as the concentration capable of neutralizing 50 or 90% of the entry efficiency (i.e., $ND_{50}$ or $ND_{90}$, respectively).

**Table 2.** X-ray data collection and structure refinement statistics (SARS-CoV-2 RBD/*Nanosota-1C* complex).

| Data collection | |
|---|---|
| Wavelength | 0.979 |
| Resolution range | 45.48–3.19 (3.30–3.19) |
| Space group | P 43 21 2 |
| Unit cell | 60.849 60.849 410.701 90 90 90 |
| Total reflections | 64167 (5703) |
| Unique reflections | 13607 (1308) |
| Multiplicity | 4.7 (4.4) |
| Completeness (%) | 96.82 (97.60) |
| Mean I/sigma(I) | 8.41 (1.80) |
| Wilson B-factor | 83.24 |
| R-merge | 0.145 (0.928) |
| R-meas | 0.1638 (1.053) |
| R-pim | 0.07385 (0.4858) |
| CC1/2 | 0.995 (0.861) |
| CC* | 0.999 (0.962) |
| **Refinement** | |
| Reflections used in refinement | 13567 (1301) |
| Reflections used for R-free | 674 (62) |
| R-work | 0.2483 (0.3521) |
| R-free | 0.2959 (0.4153) |
| CC (work) | 0.963 (0.819) |
| CC (free) | 0.909 (0.615) |
| Number of non-hydrogen atoms | 4890 |
| Macromolecules | 4833 |
| Ligands | 57 |
| Protein residues | 621 |
| RMS (bonds) | 0.002 |
| RMS (angles) | 0.45 |
| Ramachandran favored (%) | 93.11 |
| Ramachandran allowed (%) | 6.89 |
| Ramachandran outliers (%) | 0.00 |
| Rotamer outliers (%) | 3.23 |
| Clashscore | 5.25 |
| Average B-factor | 90.29 |
| Macromolecules | 89.84 |
| Ligands | 127.91 |

Statistics for the highest-resolution shell are shown in parentheses.

## SARS-CoV-2 plaque-reduction neutralization test

The potency of *Nanosota-1* in neutralizing live SARS-CoV-2 was evaluated using a SARS-CoV-2 plaque-reduction neutralization test (PRNT) assay. Specifically, the individual drug candidate was serially diluted in DMEM and mixed with SARS-CoV-2 (at a titer of 80 PFU) at 37℃ for 1 hr. The mixtures were then added into Vero E6 cells at 37℃ for an additional 45 min. After removing the culture medium, cells were overlaid with 0.6% agarose and cultured for 3 days. Plaques were visualized by

0.1% crystal violet staining. The efficacy of each drug candidate was calculated and expressed as the concentration capable of reducing the number of virus plaques by 50% (i.e., $ND_{50}$) compared to control serum-exposed virus.

## SARS-CoV-2 challenge of hamsters

Syrian hamsters (n = 24; equal sex) were obtained from Envigo (IN) and challenged via intranasal inoculation with SARS-CoV-2 ($1 \times 10^6$ $TCID_{50}$) in 100 μl DMEM (50 μl per nare). Sample size was constrained by the availability of resources. Four groups of hamsters (n = 6 in each group; randomly assigned) received *Nanosota-1C-Fc* via intraperitoneal injection at one of the following time points and dosages: (i) 24 hr pre-challenge at 20 mg/kg body weight; (ii) 4 hr post-challenge at 20 mg/kg body weight; (iii) 4 hr post-challenge at 10 mg/kg body weight. Hamsters in the negative control group were administered PBS buffer 24 hr pre-challenge. Body weights were collected daily. Hamsters were humanely euthanized on day 10 post-challenge via overexposure to $CO_2$. The lungs and bronchial tubes were collected and fixed in formalin for pathological analysis. At a sample size of six animals per group, G*Power analysis indicates that we can detect an effect size of 1.6 with a power of 0.80 (alpha = 0.05, one-tailed).

## SARS-CoV-2 challenge of human ACE2-transgenic mice

Human ACE2-transgenic mice (K18-hACE2-transgenic mice) (*Zheng et al., 2021*; *McCray et al., 2007*) (n = 35; males and females; 7–8 months old) were obtained from the Jackson Laboratories. Mice were challenged via intranasal inoculation with SARS-CoV-2 ($5 \times 10^3$ PFU) in a volume of 50 μl DMEM. Sample size was constrained by the availability of resources. Five groups of mice (n = 7 in each group) were treated with *Nanosota-1C-Fc* via intraperitoneal injection at one of the following time points and dosages: (i) 24 hr pre-challenge at 20 mg/kg body weight; (ii) 24 hr pre-challenge at 10 mg/kg body weight; (iii) 4 hr post-challenge at 20 mg/kg body weight; (iv) 4 hr post-challenge at 10 mg/kg body weight. Mice in the negative control group were administered PBS buffer 24 hr pre-challenge.

Viral titers in the lungs of mice were measured by a plaque assay. To this end, five mice from each group were euthanized on day 2 post-challenge. Lung homogenate supernatants were collected and then serially diluted in DMEM. 12-well plates of Vero E6 cells were inoculated and then incubated at 37°C in 5% $CO_2$ for 1 hr and gently rocked every 15 min. After removing the inocula, the plates were overlaid with 0.6% agarose containing 2% FBS. After 3 days, overlays were removed and plaques were visualized via staining with 0.1% crystal violet. Viral titers were quantified as PFU per ml tissue. At a sample size of five animals per group, G*Power analysis indicates that we can detect an effect size of 1.72 with a power of 0.80 (alpha = 0.05, one-tailed).

Histological examination of lungs was performed. For this pupose, the remaining two mice from each group were euthanized on day 5 post-challenge and then perfused transcardially with PBS. Mouse lungs were fixed in formalin. Sections (approximately 4 μm each) were stained with hematoxylin and eosin.

## In vivo stability of *Nanosota-1*

Male C57BL/6 mice (3–4 weeks old) (Envigo) were intravenously injected (tail vein) with *Nanosota-1C* or *Nanosota-1C-Fc* (100 μg in 100 μl PBS buffer). At varying time points, mice were euthanized and whole blood was collected. Then sera were prepared through centrifugation of the whole blood at 1500 x*g* for 10 min. The sera were then subjected to ELISA for evaluation of their SARS-CoV-2 RBD-binding capability.

## Biodistribution of *Nanosota-1* in mice

To evaluate the in vivo biodistribution of *Nanosota-1C-Fc* and *Nanosota-1C*, the nanobodies were labeled with [89Zr] and injected into male C57BL/6 mice (5–6 weeks old) (Envigo). Briefly, the nanobodies were first conjugated to the bifunctional chelator p-SCN-Bn-Deferoxamine (DFO, Macrocyclic) as previously described (*Zeglis and Lewis, 2015*). [89Zr] (University of Wisconsin Medical Physics Department) was then conjugated as previously described (*Hintz et al., 2020*). [89Zr]-labeled nanobodies (1.05 MBq, 1–2 μg nanobody, 100 μl PBS) were intravenously injected (tail vein). Mice were euthanized at various time points. Organs were collected and counted on an automatic gamma-

counter (Hidex). The total number of counts per minute (cpm) for each organ or tissue was compared with a standard sample of known activity and mass. Count data were corrected to both background and decay. The percent injected dose per gram (%ID/g) was calculated by normalization to the total amount of activity injected into each mouse.

## Acknowledgements

The development of *Nanosota-1* and animal testing were supported by funding from the University of Minnesota (to FL) and NIH grants R01AI157975 (to FL, AML, LD, SP), R01AI089728 (to FL), and R35GM118047 (to HA). AML is a 2013 Prostate Cancer Foundation Young Investigator and the recipient of a 2018 Prostate Cancer Foundation Challenge Award. Experimental Pathology Laboratories analyzed pathology data on SARS-CoV-2-challenged hamsters. Crystallization screening was performed at Hauptman-Woodward Medical Research Institute and supported by NSF grant 2029943. X-ray diffraction data were collected at Advanced Photon Source beamline 24-ID-E and supported by NIH grants P30 GM124165 and S10OD021527 and DOE contract DE-AC02-06CH11357. We thank Surajit Banerjee for help with X-ray data collection. The University of Minnesota has filed a patent on *Nanosota-1* with FL, GY, AML, JPG, JS, and YW as inventors. Coordinates and structure factors have been deposited to the Protein Data Bank with accession number 7KM5.

## Additional information

### Competing interests

Gang Ye, Joseph Gallant, Jian Shang, Yushun Wan, Aaron LeBeau, Fang Li: The University of Minnesota has filed a patent on Nanosota-1 drugs with F.L, G.Y., A.M.L., J.P.G., J.S., and Y.W. as inventors. The other authors declare that no competing interests exist.

### Funding

| Funder | Grant reference number | Author |
| --- | --- | --- |
| National Institutes of Health | R01AI157975 | Lanying Du<br>Stanley Perlman<br>Aaron LeBeau<br>Fang Li |
| National Institutes of Health | R01AI089728 | Fang Li |
| National Institutes of Health | R35GM118047 | Hideki Aihara |
| University of Minnesota | | Fang Li |
| Prostate Cancer Foundation | | Aaron LeBeau |

The funders had no role in study design, data collection and interpretation, or the decision to submit the work for publication.

### Author contributions

Gang Ye, Joseph Gallant, Conceptualization, Data curation, Formal analysis, Validation, Investigation, Visualization, Methodology, Writing - review and editing; Jian Zheng, Wanbo Tai, Data curation, Validation, Investigation, Visualization, Methodology; Christopher Massey, Ke Shi, Data curation, Validation, Investigation, Visualization, Methodology, Writing - review and editing; Abby Odle, Molly Vickers, Data curation, Investigation; Jian Shang, Yushun Wan, Conceptualization, Data curation, Validation, Investigation, Visualization, Methodology, Writing - review and editing; Lanying Du, Hideki Aihara, Funding acquisition, Validation, Investigation, Visualization, Methodology, Writing - review and editing; Stanley Perlman, Funding acquisition, Validation, Investigation, Visualization, Methodology; Aaron LeBeau, Conceptualization, Resources, Formal analysis, Supervision, Funding acquisition, Validation, Investigation, Visualization, Methodology, Project administration, Writing - review and editing; Fang Li, Conceptualization, Resources, Formal analysis, Supervision, Funding acquisition, Validation, Investigation, Visualization, Methodology, Writing - original draft, Project administration, Writing - review and editing

## Author ORCIDs

Gang Ye (iD) https://orcid.org/0000-0001-6034-2174
Joseph Gallant (iD) http://orcid.org/0000-0003-4943-1744
Wanbo Tai (iD) http://orcid.org/0000-0002-9864-8993
Hideki Aihara (iD) http://orcid.org/0000-0001-7508-6230
Fang Li (iD) https://orcid.org/0000-0002-1958-366X

## Ethics

Animal experimentation: This study was performed in strict accordance with the recommendations in the Guide for the Care and Use of Laboratory Animals of the National Institutes of Health. All of the animals were handled according to approved institutional animal care and use committee (IACUC) protocols of the University of Texas Medical Branch (protocol number 2007072), of the New York Blood Center (protocol number 194.22), of the University of Iowa (protocol number 9051795), and of the University of Minnesota (protocol number 2009-38426A).

## Decision letter and Author response

Decision letter https://doi.org/10.7554/eLife.64815.sa1
Author response https://doi.org/10.7554/eLife.64815.sa2

# Additional files

## Supplementary files

• Source data 1. Raw data for figures and figure supplements.

• Transparent reporting form

## Data availability

Coordinates and structure factors have been deposited to the Protein Data Bank with accession number 7KM5.

The following dataset was generated:

| Author(s) | Year | Dataset title | Dataset URL | Database and Identifier |
|---|---|---|---|---|
| Ye G, Shi K, Aihara H, Li F | 2021 | Crystal structure of SARS-CoV-2 RBD complexed with Nanosota-1 | https://www.rcsb.org/structure/7KM5 | RCSB Protein Data Bank, 7KM5 |

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
