## [Decision Letter]

**Acceptance summary:**

Camelid-derived nanobodies share some unique properties such as high affinity, thermal stability, ease of production etc. and can be applied as both therapeutic and diagnostic agents. This study describes a nanobody that inhibits SARS-CoV-2 and has the potential to be a drug candidate for COVID-19.

**Decision letter after peer review:**

Thank you for submitting your article "The Development of a Novel Nanobody Therapeutic for SARS-CoV-2" for consideration by *eLife*. Your article has been reviewed by 3 peer reviewers, one of whom is a member of our Board of Reviewing Editors, and the evaluation has been overseen Jos van der Meer as the Senior Editor. The following individual involved in review of your submission has agreed to reveal their identity: Zhiwei Wu (Reviewer #3).

The reviewers have discussed the reviews with one another and the Reviewing Editor has drafted this decision to help you prepare a revised submission.

Please note that the major concern for all reviewers was lack of in vivo efficacy data with respect to viral titers. This will be the most critical data needed for competitive resubmission. Each reviewer also had additional comments and suggestions to improve the text, each of which should be addressed upon revision.

Title: As noted by reviewer 2, the phrase in the title "novel nanobody therapeutic" has the potential to be misleading as several other similar nanobodies have been reported. Accordingly, we would request that the authors revise the title of any potential revision.

*Reviewer #1:*

Summary: The authors aimed to develop camelid nanobodies for treatment of SARS-CoV-2.

Strengths: The camelid nanobody developed has promising antiviral efficacy in vitro and reduced viral disease symptoms in vivo. This appears to be one of the first demonstrations of in vivo efficacy for a SARS-CoV-2 nanobody.

Weaknesses: The reduction in viral titers is not as robust as would be predicted for a highly effective antiviral antibody. It is not clear whether this nanobody would be more effective than other monoclonal antibody therapies already in use.

Appraisal: The authors have largely achieved their aim of developing a camelid nanobody that shows anti-SARS-CoV-2 efficacy.

Impact: This work may provide impetus for further development of nanobodies in human clinical settings.

1. Figure S7. The impact of the nanobody on viral titers seems quite model relative to the impact on weight loss. It would seem that a highly effective nanobody would more robustly reduce viral burden, by more than the half-log drop shown here. By comparison maybe therapy in a hamster model results in 2-3 log drop in SARS-CoV-2 viral RNA. Can the authors examine viral RNA from these samples? What else might explain why the viral load is not diminished more than it is? Also, the statistical analyses of these data are not clear. Only one p value for each group is mentioned in legend. It would seem that a multiple comparison test should be able to provide a p-value for every dose, at each time point.

2. Figure 4. It would seem appropriate to include some histology images corresponding to the clinical scoring parameter.

3. Figure 5C. Is there a negative control that can be included for comparison?

*Reviewer #2:*

In this manuscript by Ye et al., the authors employed a phage display library based on naive camelids to identify nanobodies that target the receptor-binding domain (RBD) of SARS-CoV-2. A clone named Nanosota-1A was identified from the initial screen, which was further optimized for RBD binding by in vitro affinity maturation. The resulting nanobody Nanosota-1C binds moderately the RBD at 14 nM. The authors determined the crystal structure of Nanosota-1C in complex with the RBD and found that Nanosota-1C partially overlap with the ACE2 binding sites.

To potentially enable bivalent binding and increase the pharmacological properties, the authors fused the lead construct with an Fc domain (Nanosota-1C-Fc) and positively evaluated its neutralization potential (with an NC50 of sub-microgram/ml) using a clinical isolate of the virus. Moreover, Ye and colleagues performed experiments to evaluate the in vivo efficacy of Nanosota-1C-Fc (for prophylaxis and potential treatment) in a COVID model using golden hamsters. For treatment, they evaluated two doses of a lead construct (10 mg/kg and 20 mg/kg) by intraperitoneal (IP) injection 6 hours post-infection (IN/intranasal-based inoculation). The therapeutic efficacy was evaluated based on body weight, lung pathology, and virus tilter from nasal swabs. Both prophylaxis (pre-IN) and treatment (post-IN) groups post-infection showed potential protections on body weights compared to the control group. The lead construct also appears to mitigate lung infections with decreased BAL hyperplasia, which is especially true for pre-IN animals, and less so for the post-IN(treatment) group, consistent with in vivo studies using monoclonal antibodies. Moreover, no significant decrease in nasal viral titers was detected.

The authors further expanded the study by evaluating the solubility, stability, and biodistribution of the lead construct. They provided data to indicate that Nanosota-1C-Fc can be expressed in *E. coli* with high yield, and is potentially stable both in vitro and in vivo. Using a wild type mouse model, they found that following i.v., Nanosota-1C-Fc is widely distributed in the major organs.

While the lead construct is not among the most potent nanobodies developed to date, this study is highly comprehensive including in vivo evaluations in the hamster model. Clearly, there were lots of efforts and I appreciate the amount of data presented here, from screening/ in vitro affinity maturation, structural characterization, bioengineering, biophysics to in vivo efficacy studies. However, I do have the following concerns that need to be addressed.

1. The overall flow is fine, however, the manuscript will need to be improved for clarity and conciseness. There is grammar throughout the text. Finally, the Discussion section seems to be a bit awkward and lengthy.

2. The major advantages of nanobodies over monoclonal IgG antibodies include the small size, biophysical properties for COVID, potential inhalation delivery, as well as high bioengineering potentials. In the present work, the lead construct (Nanosota-1C-Fc) was back-engineered into a full-length antibody comparable with the monoclonal antibodies directly isolated from COVID patients. The in vitro neutralization potency of the Fc fusion construct (Nanosota-1C-Fc) appears to be considerably inferior to other monoclonal COVID antibodies under clinical evaluations(e.g., single-digit nanogram/ml). Would the authors elaborate on the key conceptual advances of this work?

3. The affinity of Nanosota-1C-Fc: the authors claimed that there was a three-log improvement on the RB binding affinity, after fusing monomeric Nanosota-1C to an Fc (from 14 nM to ~ 16 pM). This raises the possibility of avidity binding. However, since the RBD does not dimerize, how could bivalent binding occur? Along the same line, I wonder if the ACE2 competition assay could be performed on Nanosota-1C-Fc to substantiate the KD measurement.

4. Line 187: the NC50 curves were plotted without sufficient data points and dilutions ( Figure 3B and S6), the quantitative differences (20 fold and 6,000 fold) therefore can not be accurately derived.

5. Literature: please expand the published literature on COVID nanobodies:

Hanke, L., Vidakovics Perez, L., Sheward, D.J., Das, H., Schulte, T., Moliner-Morro, A., Corcoran, M., Achour, A., Karlsson Hedestam, G.B., Hallberg, B.M., et al. (2020). An alpaca nanobody neutralizes SARS-CoV-2 by blocking receptor interaction. Nat Commun 11, 4420.

Xiang, Y., Nambulli, S., Xiao, Z., Liu, H., Sang, Z., Duprex, W.P., Schneidman-Duhovny, D., Zhang, C., and Shi, Y. (2020). Versatile and multivalent nanobodies efficiently neutralize SARS-CoV-2. Science 370, 1479-1484.

Schoof, M., Faust, B., Saunders, R.A., Sangwan, S., Rezelj, V., Hoppe, N., Boone, M., Billesbolle, C.B., Puchades, C., Azumaya, C.M., et al. (2020). An ultrapotent synthetic nanobody neutralizes SARS-CoV-2 by stabilizing inactive Spike. Science 370, 1473-1479.

Wrapp, D., De Vlieger, D., Corbett, K.S., Torres, G.M., Wang, N., Van Breedam, W., Roose, K., van Schie, L., Team, V.-C.C.-R., Hoffmann, M., et al. (2020). Structural Basis for Potent Neutralization of Betacoronaviruses by Single-Domain Camelid Antibodies. Cell 181, 1436-1441.

6. Regarding the weight increase of hamsters: there was evidence to indicate that weight recovery/increase in the infected hamsters is likely non-specifically related to the IP administration. What is the placebo/control used for Figure 4A? Is that PBS injection? It is not clear what "control (no drug)" indicates here. Did the author perform any post-mortem lung viral titer analysis, especially 3-4 days post-infection? This key data may provide direct evidence about how efficiently the construct can inhibit the virus in vivo.

7. Related to question 6. Since there were marginal differences (generally less than one log) of viral titers in the nasal swabs, I'd appreciate it if the author can provide more direct evidence to demonstrate the virus reductions. Minimally, examining the virus RNA levels in the lung would help substantiate the study.

8. Fc and vector construct information is missing. Is it a glycosylation variant that enables *E. coli* expression?

9. Pharmacokinetics and in vivo half-life: the authors claim that the T_1/2_ of Nanosota-1C-Fc in mice (by tail vein injection) is over 10 days. However, it is hard to conclude simply based on Figure 5C. Would it the possibility that the authors show the standard PK curve to indicate the distribution phase and to carefully derive the clearance rate?

*Reviewer #3:*

Geng Ye et al. in "The development of a Novel Nanobody Therapeutic for SARS-CoV-2" reported an isolation and characterization of nanobodies from a camelid antibody library and identified and generated Nanosota-I nanobodies for their SARS-CoV-2 inhibitory activities. The authors found that Nanosota-1C bound to both the closed and open conformations of the RBD with high affinities. The nanobody also exhibited inhibitory activity against pseudotyped SARS-CoV-2 infection of cells in vitro and inhibited live virus replication and lessened pathology in a hamster model. In addition, the authors also demonstrated that Nanosota-1C-Fc exhibited excellent stability in vivo and bioavailability, and suggest that Nanosota-1C-Fc is a potential therapeutic and preventative candidate for COVID-19. The study is well designed and executed with evidence that supports some of the conclusions.

The study demonstrated biochemically that Nanosota-1C-Fc binds both the close and open conformation of the RBD of SARS-CoV-2 and that the nanobody neutralized pseudotyped SARS-CoV-2 in in vitro assay. However, the in vivo viral inhibitory activity of the nanobody was not supported by the data since no convincing evidence was presented.

The study also suffers from incomplete investigation of the antibody role in controlling the virus since the authors did not investigate the nanobody in the nasal compartment where virus was easily detected.

Due to the unique immunological and biochemical charactertistics of the nanobodies, this study will have some impact on the therapeutic antibody development against COVID-19.

However, there remain a number of serious concerns that need to be addressed before the manuscript can be accepted for publication:

1. For the in vitro neutralization study, the authors only calculated ND50 values. The ND90 values are more meaningful biologically with respect to the efficacy of inhibiting viral infection and reflect the potency of an antibody in suppressing the virus. The authors should calculate ND90 values and presented in the text.

2. For the in vivo study, the viral load in various settings showed essentially no differences across all time points as shown in Figure S7, suggesting that the Nanosota-1C-Fc either did not inhibit the viral infection or the intravenous-administrated nanobody did not reach functional concentrations in the nasal compartment. Did the authors measure Nanosota-1C-Fc concentrations in the nasal cavity ? Figure 5D clearly indicated that there is little antibody in the lungs.

3. In view of the Figure S7 data, did the authors measure viral titers in other compartments, particularly lungs, to see if virus was inhibited by Nanosota-1C-Fc. It is critical that the mitigation of the pathology is indeed caused by the reduced viral infection/replication rather than other mechanisms.

4. Based on the data presented, there is no evidence that the nanobody inhibited virus in the in vivo model. How did the administration of Nanosota-1C-Fc resulted in the reduction of disease severity ? The authors need to provide an explanation.

5. SARS-CoV-2 infection induces marked inflammatory cytokine secretion which causes extensive organ damage and is the major pathogenic mechanisms of the viral infection. Did the authors look into the inflammatory cytokine production before and after Nanosota-1C-Fc administration ? Using bronchioloalveolar hyperplasia alone is not sufficient to indicate the efficacy of the antibody.

6. The authors overstated the roles of Nanosota-1C-Fc as a protective agent against SARS-CoV-2 infection. With 1-2 weeks of in vivo life, using antibody for prevention is both costly and ineffective. In particular, if Nanosota-1C-Fc is not present in the nasal compartment or oral/upper respiratory mucosa in functional concentrations, intravenously administrated Nanosota-1C-Fc will not prevent the viral transmission. I suggest that the authors remove these statements.

[Editors' note: further revisions were suggested prior to acceptance, as described below.]

Thank you for resubmitting your work entitled "The Development of *Nanosota-1*as anti-SARS-CoV-2 nanobody drug candidates" for further consideration by *eLife*. Your revised article has been evaluated by Jos van der Meer as the Senior Editor, and a Reviewing Editor.

The manuscript has been improved but there are some remaining issues that need to be addressed, as outlined below:

1. At least two published studies have evaluated the preclinical efficacy in rodents, including innovative use of nanobody aerosolization for inhalation therapy of SARS-CoV-2 infection.

i) Nanobody cocktails potently neutralize SARS-CoV-2 D614G N501Y variant and protect mice

ii) Inhalable Nanobody (PiN-21) prevents and treats SARS-CoV-2 infections in Syrian hamsters at ultra-low doses.

These studies should be discussed in the context of your work.

2. Since Nanosota targets highly variable ACE2 binding sites, it may not resist many circulating VOCs (especially E484K/Q that shares with beta and the prevalent delta variants). It would be useful to discuss this vis-a-vis the evolving virus.

---

## [Author Response]

Please note that the major concern for all reviewers was lack of in vivo efficacy data with respect to viral titers. This will be the most critical data needed for competitive resubmission. Each reviewer also had additional comments and suggestions to improve the text, each of which should be addressed upon revision.

Summary of new data collections:

1) To further examine the in vivo efficacy of *Nanosota-1C-Fc*, we evaluated its therapeutic efficacy in a human ACE2-transgenic mouse model challenged with SARS-CoV-2. We measured the virus titer in the mouse lungs and also took photos of the lung histology (Figure 4C; Figure 4D). The results showed that compared to the untreated control group, the mice that received *Nanosota-1C-Fc* had much lower virus titers in the lungs (2~3 logs lowers) and showed improvement in the lung pathology. Combined with the data from the hamster study, the current manuscript showed that *Nanosota-1C-Fc* demonstrated preventive and therapeutic efficacy against live SARS-CoV-2 infection in hamster and mouse models.

2) We repeated the protein pull-down assay, with *Nanosota-1C-Fc* replacing *Nanosota-1C* (Figure 3—figure supplement 3B). The result confirmed that *Nanosota-1C-Fc* and ACE2 bound competitively to the RBD; it further showed that *Nanosota-1C-Fc* bound to the RBD much more strongly than ACE2 did, consistent with the binding affinity measurement.

Reviewer #1:[…] 1. Figure S7. The impact of the nanobody on viral titers seems quite model relative to the impact on weight loss. It would seem that a highly effective nanobody would more robustly reduce viral burden, by more than the half-log drop shown here. By comparison maybe therapy in a hamster model results in 2-3 log drop in SARS-CoV-2 viral RNA. Can the authors examine viral RNA from these samples? What else might explain why the viral load is not diminished more than it is? Also, the statistical analyses of these data are not clear. Only one p value for each group is mentioned in legend. It would seem that a multiple comparison test should be able to provide a p-value for every dose, at each time point.

We recently learned that nasal wash, which we used to assess the hamsters, is a poor source for detecting viral titers in hamsters using RT-PCR. This is based on Zhou et al. 2021 (Cell Host and Microbe 29, 551–563). The RT-PCR results of that study showed that in hamsters receiving neutralizing antibodies and challenged by SARS-CoV-2, “viral RNA copy numbers were reduced in the lungs by an average of 3 logs (range, 0.7–4.5) (Figure 3C). In contrast, there was no significant viral load reduction in nasal turbinates and trachea in both dose groups”. Because nasal wash is not a good source, the data are uninformative and not included in the revised manuscript.

A superior index of viral titers in treated animals is to measure virus titers in the animal lungs during the peak of viral infections. Although limited resources prevented us from repeating the hamster study, we were able to conduct this critical test in a mouse model. Specifically, to further examine the in vivo efficacy of *Nanosota-1C-Fc*, we evaluated its therapeutic efficacy in a human ACE2-transgenic mouse model challenged with SARS-CoV-2. Instead of monitoring the body weights of the mice through the viral infection and recovery process, we measured the virus titers in the lungs at the peak of the viral infection. To this end, four experimental groups of mice (seven per group) were injected with a single dose of *Nanosota-1C-Fc.* Five out of the seven mice from each group were euthanized on day 2 post-challenge, and the virus titers in their lungs were measured using a virus titer plaque assay (Figure 4C). The remaining two mice in each group were euthanized on day 5 post-challenge for pathology analysis of their lung tissues (Figure 4D). Compared to the untreated control group, the mice that received *Nanosota-1C-Fc* had much lower virus titers in the lungs (2-3 logs lower) and showed improvement in their lung pathology. Combined with the weight and pathology data from the hamster study, these findings showed that *Nanosota-1C-Fc* demonstrated preventive and therapeutic efficacy against live SARS-CoV-2 infection in both hamster and mouse models.

2. Figure 4. It would seem appropriate to include some histology images corresponding to the clinical scoring parameter.

The initial histology experiment on the hamsters was done by a company that didn’t provide the histology images despite our repeated requests. Instead, we collected histology images on the mouse lungs. These histology images have been added as Figure 4D in the revised manuscript.

3. Figure 5C. Is there a negative control that can be included for comparison?

PBS buffer was used as a negative control for the in vivo stability experiment. The data have been added as Figure 5—figure supplement 1B in the revised manuscript.

Reviewer #2:[…] 1. The overall flow is fine, however, the manuscript will need to be improved for clarity and conciseness. There is grammar throughout the text. Finally, the Discussion section seems to be a bit awkward and lengthy.

We shortened the discussion and carefully edited the manuscript to increase its clarity and to minimize grammatical errors.

2. The major advantages of nanobodies over monoclonal IgG antibodies include the small size, biophysical properties for COVID, potential inhalation delivery, as well as high bioengineering potentials. In the present work, the lead construct (Nanosota-1C-Fc) was back-engineered into a full-length antibody comparable with the monoclonal antibodies directly isolated from COVID patients. The in vitro neutralization potency of the Fc fusion construct (Nanosota-1C-Fc) appears to be considerably inferior to other monoclonal COVID antibodies under clinical evaluations(e.g., single-digit nanogram/ml). Would the authors elaborate on the key conceptual advances of this work?

Regarding the size of the Nanosota-1C-Fc, we added the following discussion:

“It is worth noting that the molecular weight of *Nanosota-1C-Fc* (78 kDa) is above the kidney clearance threshold (60 kDa) (8), but still only half of conventional antibodies’ molecular weight (150 kDa). […] All of these features are critical for the implementation of *Nanosota-1C-F*c as a potential COVID-19 therapeutic.”

Regarding the comparison between nanobodies and conventional antibodies, we included the following discussions:

“Unlike conventional antibodies, *Nanosota-1C-Fc* was produced at high yields in bacteria and had exceptional thermostability.”

“The widespread of COVID-19 in the world calls for large-scale manufacturing of anti-SARSCoV-2 therapeutics. Molecules such as *Nanosota-1*, with its high production yields from bacteria and excellent in vitro and in vivo stabilities, are promising drug candidates to meet this need.”

Regarding the key conceptual advances of this work, we added the following discussions:

“Nanobody therapeutics derived from camelid antibodies have advantages relative to conventional antibodies. […] The extensive scope of the work makes the current study among the most comprehensive on anti-SARS-CoV-2 nanobody drug candidates.”

3. The affinity of Nanosota-1C-Fc: the authors claimed that there was a three-log improvement on the RB binding affinity, after fusing monomeric Nanosota-1C to an Fc (from 14 nM to ~ 16 pM). This raises the possibility of avidity binding. However, since the RBD does not dimerize, how could bivalent binding occur? Along the same line, I wonder if the ACE2 competition assay could be performed on Nanosota-1C-Fc to substantiate the KD measurement.

The binding affinities were measured using surface plasmon resonance. All of the measurement curves were shown in Figure 2—figure supplement 3.

To further characterize the binding interactions between Nanosota-1C-Fc and RBD, we repeated the protein pull-down assay, with *Nanosota-1C-Fc* replacing *Nanosota-1C* (Figure 3—figure supplement 3B). The result confirmed that *Nanosota-1C-Fc* and ACE2 bound competitively to the RBD; it further showed that *Nanosota-1C-Fc* bound to the RBD much more strongly than ACE2 did, consistent with the binding affinity measurement.

4. Line 187: the NC50 curves were plotted without sufficient data points and dilutions ( Figure 3B and S6), the quantitative differences (20 fold and 6,000 fold) therefore cannot be accurately derived.

In the revised manuscript, we no longer compare the Nanosota-1C-Fc and ACE2 in their capability to neutralize live SARS-CoV-2 infections. Instead, we focus on their differences in neutralizing pseudotyped SARS-CoV-2 entry, where sufficient data points were acquired.

5. Literature: please expand the published literature on COVID nanobodies:Hanke, L., Vidakovics Perez, L., Sheward, D.J., Das, H., Schulte, T., Moliner-Morro, A., Corcoran, M., Achour, A., Karlsson Hedestam, G.B., Hallberg, B.M., et al. (2020). An alpaca nanobody neutralizes SARS-CoV-2 by blocking receptor interaction. Nat Commun 11, 4420.Xiang, Y., Nambulli, S., Xiao, Z., Liu, H., Sang, Z., Duprex, W.P., Schneidman-Duhovny, D., Zhang, C., and Shi, Y. (2020). Versatile and multivalent nanobodies efficiently neutralize SARS-CoV-2. Science 370, 1479-1484.Schoof, M., Faust, B., Saunders, R.A., Sangwan, S., Rezelj, V., Hoppe, N., Boone, M., Billesbolle, C.B., Puchades, C., Azumaya, C.M., et al. (2020). An ultrapotent synthetic nanobody neutralizes SARS-CoV-2 by stabilizing inactive Spike. Science 370, 1473-1479.Wrapp, D., De Vlieger, D., Corbett, K.S., Torres, G.M., Wang, N., Van Breedam, W., Roose, K., van Schie, L., Team, V.-C.C.-R., Hoffmann, M., et al. (2020). Structural Basis for Potent Neutralization of Betacoronaviruses by Single-Domain Camelid Antibodies. Cell 181, 1436-1441.

We have included these references in the revised manuscript.

6. Regarding the weight increase of hamsters: there was evidence to indicate that weight recovery/increase in the infected hamsters is likely non-specifically related to the IP administration. What is the placebo/control used for Figure 4A? Is that PBS injection? It is not clear what "control (no drug)" indicates here. Did the author perform any post-mortem lung viral titer analysis, especially 3-4 days post-infection? This key data may provide direct evidence about how efficiently the construct can inhibit the virus in vivo.

In both the figure legend and methods, we have clarified that PBS buffer was the negative control in animal experiments.

A superior index of viral titers in treated animals is to measure virus titers in the animal lungs during the peak of viral infections. Although limited resources prevented us from repeating the hamster study, we were able to conduct this critical test in a mouse model. Specifically, to further examine the in vivo efficacy of *Nanosota-1C-Fc*, we evaluated its therapeutic efficacy in a human ACE2-transgenic mouse model challenged with SARS-CoV-2. Instead of monitoring the body weights of the mice through the viral infection and recovery process, we measured the virus titers in the lungs at the peak of the viral infection. To this end, four experimental groups of mice (seven per group) were injected with a single dose of *Nanosota-1C-Fc.* Five out of the seven mice from each group were euthanized on day 2 post-challenge, and the virus titers in their lungs were measured using a virus titer plaque assay (Figure 4C). The remaining two mice in each group were euthanized on day 5 post-challenge for pathology analysis of their lung tissues (Figure 4D). Compared to the untreated control group, the mice that received *Nanosota-1C-Fc* had much lower virus titers in the lungs (2-3 logs lower) and showed improvement in their lung pathology. Combined with the weight and pathology data from the hamster study, these findings showed that *Nanosota-1C-Fc* demonstrated preventive and therapeutic efficacy against live SARS-CoV-2 infection in both hamster and mouse models.

7. Related to question 6. Since there were marginal differences (generally less than one log) of viral titers in the nasal swabs, I'd appreciate it if the author can provide more direct evidence to demonstrate the virus reductions. Minimally, examining the virus RNA levels in the lung would help substantiate the study.

We recently learned that nasal wash, which we used to assess the hamsters, is a poor source for detecting viral titers in hamsters using RT-PCR. This is based on Zhou et al. 2021 (Cell Host and Microbe 29, 551–563). The RT-PCR results of that study showed that in hamsters receiving neutralizing antibodies and challenged by SARS-CoV-2, “viral RNA copy numbers were reduced in the lungs by an average of 3 logs (range, 0.7–4.5) (Figure 3C). In contrast, there was no significant viral load reduction in nasal turbinates and trachea in both dose groups”. Because nasal wash is not a good source, the data are uninformative and not included in the revised manuscript.

We undertook an important step by measuring virus titers in the lungs in a mouse model. The data did provide more direct evidence for virus reductions after treatment. Please see our response to point #6.

8. Fc and vector construct information is missing. Is it a glycosylation variant that enables *E. coli* expression?

We included Fc and vector construct information in the Methods:

“*Nanosota-1C-Fc* was cloned into pET42b vector (Novagen) with a C-terminal human IgG_1_ Fc tag.”

We didn’t mutate the glycosylation site in the Fc tag.

9. Pharmacokinetics and in vivo half-life: the authors claim that the T_1/2_ of Nanosota-1C-Fc in mice (by tail vein injection) is over 10 days. However, it is hard to conclude simply based on Figure 5C. Would it the possibility that the authors show the standard PK curve to indicate the distribution phase and to carefully derive the clearance rate?

In the revised manuscript, we have removed discussions on the half-life of Nanosota1C-Fc. Instead, we only discuss “its excellent in vivo stability (significant RBD-binding capacity remained after 10 days in vivo).”

Reviewer #3:[…] There remain a number of serious concerns that need to be addressed before the manuscript can be accepted for publication:1. For the in vitro neutralization study, the authors only calculated ND50 values. The ND90 values are more meaningful biologically with respect to the efficacy of inhibiting viral infection and reflect the potency of an antibody in suppressing the virus. The authors should calculate ND90 values and presented in the text.

In the revision, we calculated ND90 for Nanosota-1C-Fc and Nanosota-1C in neutralizing SARS-CoV-2 pseudovirus entry and presented them in both the text and Figure 3A/ Figure 3—figure supplement 1.

We didn’t calculate ND90 for the proteins in neutralizing live SARS-CoV-2 infection because, as Reviewer 2 pointed out, the data points in these experiments were insufficient for reliable calculations.

2. For the in vivo study, the viral load in various settings showed essentially no differences across all time points as shown in Figure S7, suggesting that the Nanosota-1C-Fc either did not inhibit the viral infection or the intravenous-administrated nanobody did not reach functional concentrations in the nasal compartment. Did the authors measure Nanosota-1C-Fc concentrations in the nasal cavity ? Figure 5D clearly indicated that there is little antibody in the lungs.3. In view of the Figure S7 data, did the authors measure viral titers in other compartments, particularly lungs, to see if virus was inhibited by Nanosota-1C-Fc. It is critical that the mitigation of the pathology is indeed caused by the reduced viral infection/replication rather than other mechanisms.4. Based on the data presented, there is no evidence that the nanobody inhibited virus in the in vivo model. How did the administration of Nanosota-1C-Fc resulted in the reduction of disease severity ? The authors need to provide an explanation.

We recently learned that nasal wash, which we used to assess the hamsters, is a poor source for detecting viral titers in hamsters using RT-PCR. This is based on Zhou et al. 2021 (Cell Host and Microbe 29, 551–563). The RT-PCR results of that study showed that in hamsters receiving neutralizing antibodies and challenged by SARS-CoV-2, “viral RNA copy numbers were reduced in the lungs by an average of 3 logs (range, 0.7–4.5) (Figure 3C). In contrast, there was no significant viral load reduction in nasal turbinates and trachea in both dose groups”. Because nasal wash is not a good source, the data are uninformative and not included in the revised manuscript.

A superior index of viral titers in treated animals is to measure virus titers in the animal lungs during the peak of viral infections. Although limited resources prevented us from repeating the hamster study, we were able to conduct this critical test in a mouse model. Specifically, to further examine the in vivo efficacy of *Nanosota-1C-Fc*, we evaluated its therapeutic efficacy in a human ACE2-transgenic mouse model challenged with SARS-CoV-2. Instead of monitoring the body weights of the mice through the viral infection and recovery process, we measured the virus titers in the lungs at the peak of the viral infection. To this end, four experimental groups of mice (seven per group) were injected with a single dose of *Nanosota-1C-Fc.* Five out of the seven mice from each group were euthanized on day 2 post-challenge, and the virus titers in their lungs were measured using a virus titer plaque assay (Figure 4C). The remaining two mice in each group were euthanized on day 5 post-challenge for pathology analysis of their lung tissues (Figure 4D). Compared to the untreated control group, the mice that received *Nanosota-1C-Fc* had much lower virus titers in the lungs (2-3 logs lower) and showed improvement in their lung pathology. Combined with the weight and pathology data from the hamster study, these findings showed that *Nanosota-1C-Fc* demonstrated preventive and therapeutic efficacy against live SARS-CoV-2 infection in both hamster and mouse models.

5. SARS-CoV-2 infection induces marked inflammatory cytokine secretion which causes extensive organ damage and is the major pathogenic mechanisms of the viral infection. Did the authors look into the inflammatory cytokine production before and after Nanosota-1C-Fc administration ? Using bronchioloalveolar hyperplasia alone is not sufficient to indicate the efficacy of the antibody.

In our human ACE2 transgenic model challenged by SARS-CoV-2, most cytokine/chemokine increase occurred at 4 and 6 dpi. (J. Zheng et al., COVID-19 treatments and pathogenesis including anosmia in K18-hACE2 mice. Nature 589, 603-607 (2021).). In the current study, lungs of five mice from each group were collected at 2 dpi for the purpose of measuring virus titers in lungs. Consistent with our previously published results, for samples collected at this time point, we didn’t find a significant increase in cytokine secretion even in the PBS control group. Nevertheless, in the revised manuscript, we showed that compared to the untreated control group, the mice that received *Nanosota-1C-Fc* had much lower virus titers in the lungs (2-3 logs lower) and showed improvement in their lung pathology. Combined with the data from the hamster study, the current manuscript showed that *Nanosota-1C-Fc* demonstrated preventive and therapeutic efficacy against live SARS-CoV-2 infection in both hamster and mouse models.

6. The authors overstated the roles of Nanosota-1C-Fc as a protective agent against SARS-CoV-2 infection. With 1-2 weeks of in vivo life, using antibody for prevention is both costly and ineffective. In particular, if Nanosota-1C-Fc is not present in the nasal compartment or oral/upper respiratory mucosa in functional concentrations, intravenously administrated Nanosota-1C-Fc will not prevent the viral transmission. I suggest that the authors remove these statements.

We removed these statements and added the following discussion:

“It is worth noting that given the recent success of COVID-19 vaccines, future development of *Nanosota-1* series should focus on its therapeutic efficacy rather than its preventive efficacy.”

[Editors' note: further revisions were suggested prior to acceptance, as described below.]

The manuscript has been improved but there are some remaining issues that need to be addressed, as outlined below:1. At least two published studies have evaluated the preclinical efficacy in rodents, including innovative use of nanobody aerosolization for inhalation therapy of SARS-CoV-2 infection.i) Nanobody cocktails potently neutralize SARS-CoV-2 D614G N501Y variant and protect miceii) Inhalable Nanobody (PiN-21) prevents and treats SARS-CoV-2 infections in Syrian hamsters at ultra-low doses.These studies should be discussed in the context of your work.

In the revised manuscript, we have revised the discussion as follows:

“Although several studies have reported nanobody drug candidates that specifically target SARS-CoV-2 (12-16, 32, 33), as of this writing, only two studies have evaluated their nanobody drug candidates in an animal model for anti-SARS-CoV-2 therapeutic efficacy (32, 33). None of these studies have evaluated their nanobody drug candidates for in vitro thermostability, in vivo stability, or tissue biodistribution.”

We also added one new reference (in red) to the following discussion:

“Third, despite its rapid clearance from the blood, *Nanosota-1C* could be used as an inhaler to treat infections in the respiratory tracts (10, 33) or as an oral drug to treat infections in the intestines (11).”

2. Since Nanosota targets highly variable ACE2 binding sites, it may not resist many circulating VOCs (especially E484K/Q that shares with beta and the prevalent delta variants). It would be useful to discuss this vis-a-vis the evolving virus.

In the revised manuscript, we have added the following discussion:

“Moreover, to combat emerging SARS-CoV-2 variants with RBM mutations, *Nanosota-1* series can be further developed through additional affinity maturation against the RBD of SARS-CoV2 variants. The series may also be developed to become part of cocktail therapies that target multiple sites on SARS-CoV-2.”